# A Gradient Guided Diffusion Framework for Chance Constrained Programming

**Boyang Zhang**
School of Advanced Interdisciplinary Sciences
University of Chinese Academy of Sciences
Beijing 100049, China
zhangboyang23@mails.ucas.ac.cn

**Zhiguo Wang**[*]
Department of Mathematics
Sichuan University
Chengdu 610065, China
wangzhiguo@scu.edu.cn

**Ya-Feng Liu**[*]
Ministry of Education Key Laboratory of Mathematics and Information Networks
School of Mathematical Sciences
Beijing University of Posts and Telecommunications
Beijing 102206, China
yafengliu@bupt.edu.cn

## Abstract

Chance constrained programming (CCP) is a powerful framework for addressing optimization problems under uncertainty. In this paper, we introduce a novel **G**radient-**G**uided **D**iffusion-based **Opt**imization framework, termed GGDOpt, which tackles CCP through three key innovations. First, GGDOpt accommodates a broad class of CCP problems without requiring the knowledge of the exact distribution of uncertainty—relying solely on a set of samples. Second, to address the nonconvexity of the chance constraints, it reformulates the CCP as a sampling problem over the product of two distributions: an unknown data distribution supported on a nonconvex set and a Boltzmann distribution defined by the objective function, which fully leverages both first- and second-order gradient information. Third, GGDOpt has theoretical convergence guarantees and provides practical error bounds under mild assumptions. By progressively injecting noise during the forward diffusion process to convexify the nonconvex feasible region, GGDOpt enables guided reverse sampling to generate asymptotically optimal solutions. Experimental results on synthetic datasets and a waveform design task in wireless communications demonstrate that GGDOpt outperforms existing methods in both solution quality and stability with nearly 80% overhead reduction.

Our code is available at https://github.com/boyangzhang2000/GGDOpt.

## 1 Introduction

### 1.1 Problem formulation

Chance constrained programming (CCP) is an efficient modeling paradigm for optimization problems with uncertain constraints, which finds wide applications in diverse fields, such as finance (Bonami and Lejeune [2009]), robot control (Calafiore and Campi [2006]), and wireless communications

---

[*]Corresponding authors.

(Wang et al. [2014]). In this paper, we consider a CCP with the following form:

$$\min_{\boldsymbol{x}} \quad f(\boldsymbol{x})$$
$$\text{s.t.} \quad \boldsymbol{x} \in \mathcal{X}_\rho, \tag{1}$$

where $f : \mathbb{R}^n \to \mathbb{R}$ is a differentiable objective function and $\mathcal{X}_\rho$ is the chance (or probabilistic) constraint set defined by

$$\mathcal{X}_\rho = \left\{ \boldsymbol{x} \in \mathbb{R}^n \mid \text{Prob}_{\boldsymbol{h}}\{\boldsymbol{g}(\boldsymbol{x}, \boldsymbol{h}) \geq \boldsymbol{0}\} \geq 1 - \rho \right\}. \tag{2}$$

In the above, $\boldsymbol{h}$ is a random vector with probability distribution $P$ supported on a set $\Xi \subset \mathbb{R}^d$, $\rho \in (0, 1)$, $\boldsymbol{g} = (g_1, g_2, \ldots, g_m) : \mathbb{R}^n \times \Xi \to \mathbb{R}^m$, and $\text{Prob}(A)$ denotes the probability of an event $A$. Problem (1) is generally challenging to solve for the following two reasons. First, evaluating the probability term $\text{Prob}_{\boldsymbol{h}}\{\boldsymbol{g}(\boldsymbol{x}, \boldsymbol{h}) \geq 0\}$ typically involves a high-dimensional integration, which is computationally intractable. Second, even when $\boldsymbol{g}$ is linear, the feasible set $\mathcal{X}_\rho$ remains nonconvex, further complicating the optimization.

## 1.2 Related works

Apart from very special cases where $\mathcal{X}_\rho$ can be transformed into a convex formulation under strong assumptions (Kataoka [1963], Lagoa et al. [2005], Henrion [2007], Prékopa [2013]), there are two popular approaches to tackling general problem (1), which are Convex Approximation (CA) method and Sample Average Approximation (SAA) method. The CA method seeks to construct a tractable inner approximation of $\mathcal{X}_\rho$, but it typically requires the information of the *exact* distribution $P$, often assuming that $P$ belongs to specific families such as Gaussian or log-concave distributions (Ben-Tal and Nemirovski [2000], Bertsimas and Sim [2004], Lagoa et al. [2005], Nemirovski and Shapiro [2007]). In contrast, the SAA method approximates $P$ using an empirical distribution based on sampled data, reformulating the CCP as a binary integer program (Ahmed and Shapiro [2008], Pagnoncelli et al. [2009], Adam and Branda [2016]). However, this reformulation remains computationally intractable. These restrictive assumptions on the underlying distribution $P$, along with the high computational cost, significantly limit the practical applicability of CCP.

One important question to ask is: **can we design a general framework to efficiently solve CCP when the underlying distribution $P$ is unknown?** The answer to the above question is particularly crucial in our interested case where samples can be efficiently drawn from $\mathcal{X}_\rho$, albeit the explicit formulation of $\mathcal{X}_\rho$ is unavailable. This motivates us to seek high-quality solutions to the CCP problem (1) from a new perspective via sampling-based methods (Wibisono [2018], Ma et al. [2019], Lee et al. [2021], Chen et al. [2022], Seyoum and You [2025]). The core idea of applying sampling-based methods to solve CCP problems lies in reformulating the original nonconvex CCP with intractable constraints as a sampling problem from an unknown distribution. This reformulation leverages probabilistic techniques to handle the challenging constraints through stochastic sampling rather than deterministic evaluation.

Notably, generative models are designed to approximate unknown data distributions based on observed samples, enabling the generation of new data points from the learned approximation. In particular, diffusion models have emerged as a powerful family of generative models, offering high-quality sample generation, stable training dynamics, and scalability to high-dimensional problems (Ho et al. [2020]). The sampling process based on score estimation enables diffusion models to generalize to conditional distributions, thereby generating samples that satisfy requirements through conditional information guidance (Ho and Salimans [2022]). As a powerful generative artificial intelligence (AI) technology, diffusion model has been successfully deployed across various domains, such as, image generation ( Yue et al. [2023], Huang et al. [2025]), inverse problems (Chung et al. [2022b], Chung et al. [2022a], Song et al. [2023]), and optimization (Krishnamoorthy et al. [2023], Li et al. [2024], Wu et al. [2024], Kong et al. [2024], Liang et al. [2025]). Recently, Guo et al. [2024] introduced a novel form of gradient guidance to adapt pre-trained diffusion models for user-specified tasks.

Despite their success in various domains, diffusion models have rarely been explored in the context of CCP. The possible reason behind might be that tackling CCP problems via diffusion models generally requires efficient sampling from a composite distribution, the product of an unknown data distribution (associated with the constraint) and a known Boltzmann distribution (induced by the objective function), but the training data is only available from the unknown component. This makes the application of diffusion models to CCP both novel and nontrivial.

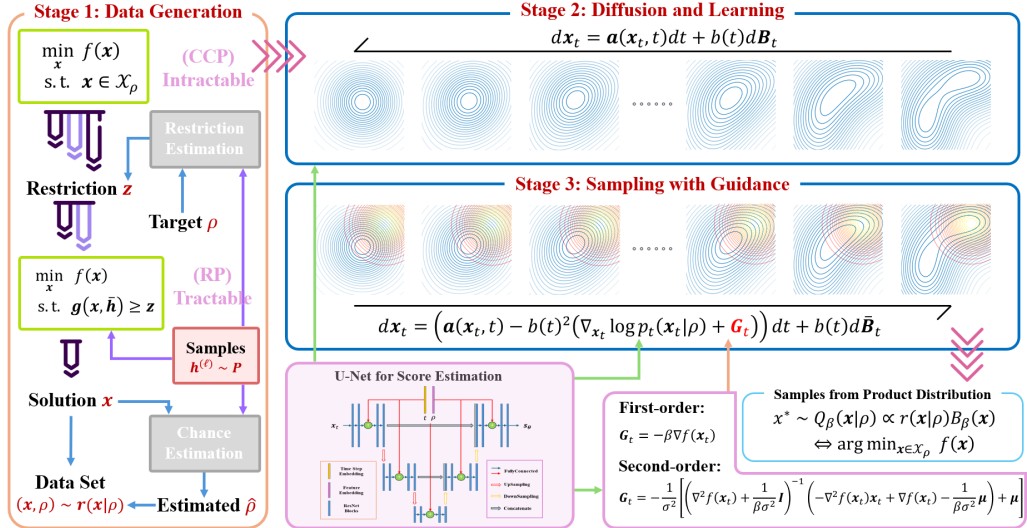

Figure 1: A framework of GGDOpt. (1) Generate a training set of points satisfying the chance constraint by solving a deterministic restricted problems. (2) Train a diffusion model with classifier-free guidance to learn the score of the conditional distribution. (3) Perform the reverse diffusion process with additional gradient guidance to sample from the product of the data distribution and the Boltzmann distribution.

## 1.3 Our contributions

In this paper, we propose GGDOpt (see Figure 1), a novel **G**radient-**G**uided **D**iffusion-based **Opt**imization framework for solving problem (1), with the following originality:

- **Applicable to broader problem domains.** Built on the basis of diffusion model with classifier-free guidance and optimization via sampling, GGDOpt accommodates a broad class of CCP problems without requiring the knowledge of the exact distribution of uncertainty—relying solely on a set of samples.

- **Problem reformulation with a novel paradigm.** GGDOpt reformulates the CCP problem as a sampling task over the product of two distributions: an unknown data distribution implicitly defined by the constraint and a Boltzmann distribution induced by the objective function with a full utilization of first- and second-order information of the underlying CCP.

- **Feasibility-aware data generation and efficient guided sampling.** To generate high-quality training data that satisfy the chance constraint, GGDOpt solves a deterministic restricted problems by standard optimization techniques. The solutions are used to guide the training of the conditional diffusion model, effectively capturing the geometry of the feasible region. To sample from the product distribution, we develop a gradient-guided reverse process derived in closed form based on the structure of the product distribution. Compared with Guo et al. [2024], our guidance terms do not require backpropagation through the neural network.

- **Theoretical convergence and practical evaluation.** Regarding the sampling process as a reverse time stochastic differential equation (SDE), GGDOpt is shown to generate asymptotically optimal solutions as the time step and inverse temperature go to infinity. A practical error bound is also provided with two components: the limited time length error and limited inverse temperature error.

## 1.4 Organization

The remainder of the paper is organized as follows. In Section 2, a reformulation of CCP problem (1) is provided via sampling, and a gradient guidance-based score estimation schedule is provided with both first- and second-order information. A novel GGDOpt framework for solving problem (1) is given in Section 3. Theoretical convergence and experimental results are presented in Section 4 and Section 5, respectively. The conclusion is drawn in Section 6.

## 2 Problem reformulation via sampling

Let $r(\boldsymbol{x}|\rho) = \mathbb{I}_{\mathcal{X}_\rho}(\boldsymbol{x})$ denote the indicator function of the chance constraint $\mathcal{X}_\rho$. Let $B_\beta(\boldsymbol{x}) \propto e^{-\beta f(\boldsymbol{x})}$ represent the Boltzmann distribution associated with the objective function $f(\boldsymbol{x})$, where $\beta > 0$. The resulting sampling task is to draw samples from the following target distribution:

$$\text{sample } \boldsymbol{x} \sim Q_\beta(\boldsymbol{x}|\rho) \propto r(\boldsymbol{x}|\rho)B_\beta(\boldsymbol{x}). \tag{3}$$

Intuitively, the distribution $Q_\beta(\boldsymbol{x}|\rho)$ assigns higher probability density to regions where the objective function $f(\boldsymbol{x})$ takes smaller values. Under certain regularity conditions (Kong et al. [2024]), as $\beta \to \infty$, the sampling distribution $Q_\beta(\boldsymbol{x}|\rho)$ asymptotically concentrates around the global minimizer of the CCP in (1). Therefore, the CCP (1) admits the following equivalent reformulation:

$$\boldsymbol{x}^* = \arg\min_{\boldsymbol{x}} \left\{ f(\boldsymbol{x}) + \mathbb{I}_{\mathcal{X}_\rho}(\boldsymbol{x}) \right\} \iff \text{sample } \boldsymbol{x}^* \sim Q_\beta(\boldsymbol{x}|\rho),\ \beta \to \infty. \tag{4}$$

A natural way would be to directly employ Langevin dynamics for sampling from distribution $Q_\beta(\boldsymbol{x}|\rho)$. However, the unknown nature of component $r(\boldsymbol{x}|\rho)$ prevents the derivation of an exact expression of the score function. Fortunately, we can obtain a set of feasible samples $\{\boldsymbol{x}^{(i)}, \rho^{(i)}\}_{i=1}^N$, which are drawn from the unknown distribution $r(\boldsymbol{x}|\rho)$. More details on this will be presented in Subsection 3.1. This motivates us to leverage diffusion models to directly learn the product distribution $Q_\beta(\boldsymbol{x}|\rho) \propto r(\boldsymbol{x}|\rho)B_\beta(\boldsymbol{x})$, where $r(\boldsymbol{x}|\rho)$ is unknown but $B_\beta(\boldsymbol{x})$ is explicitly known.

### 2.1 Diffusion models

Given observed samples $\boldsymbol{x}_0$ from a distribution of interest, the goal of a diffusion model is to learn to model its true data distribution $p_0(\boldsymbol{x}_0)$. Once learned, we can generate new samples from our approximate model at will. The diffusion model builds a diffusion process by defining a forward SDE starting from $p_0(\boldsymbol{x}_0)$ as follows:

$$d\boldsymbol{x}_t = \boldsymbol{a}(\boldsymbol{x}_t, t)dt + b(t)d\boldsymbol{B}_t, \tag{5}$$

where $t \in [0, T]$, $\boldsymbol{B}_t$ is the standard Wiener process (a.k.a., Brownian motion), $\boldsymbol{a}(\cdot, t) : \mathbb{R}^d \to \mathbb{R}^d$ is a vector valued function called the drift coefficient, and $b(\cdot) : \mathbb{R} \to \mathbb{R}$ is a scalar function known as the diffusion coefficient.

By starting from samples of $\boldsymbol{x}_T \sim p_T(\boldsymbol{x}_T)$ and reversing the process, we can obtain samples $\boldsymbol{x}_0 \sim p_0(\boldsymbol{x}_0)$. The reverse of a diffusion process is also a diffusion process, running backwards in time and given by the following reverse-time SDE:

$$d\boldsymbol{x}_t = \left( \boldsymbol{a}(\boldsymbol{x}_t, t) - b(t)^2 \nabla_{\boldsymbol{x}_t} \log p_t(\boldsymbol{x}_t) \right) dt + b(t)d\bar{\boldsymbol{B}}_t, \tag{6}$$

where $\bar{\boldsymbol{B}}_t$ is a standard Wiener process when the time flows backwards from $T$ to 0. The only unknown term $\nabla_{\boldsymbol{x}_t} \log p_t(\boldsymbol{x}_t)$ is the score function of the marginal density $p_t(\boldsymbol{x}_t)$.

To estimate $\nabla_{\boldsymbol{x}_t} \log p_t(\boldsymbol{x}_t)$, we can train a time-dependent score-based model $\boldsymbol{s}_{\boldsymbol{\theta}}(\boldsymbol{x}_t, t)$ with

$$\boldsymbol{\theta}^* = \arg\min_{\boldsymbol{\theta}} \mathbb{E}_{t \sim \mathcal{U}[0,T]} \left\{ \lambda_t \mathbb{E}_{\boldsymbol{x}_0} \mathbb{E}_{\boldsymbol{x}_t|\boldsymbol{x}_0} \left[ \|\boldsymbol{s}_{\boldsymbol{\theta}}(\boldsymbol{x}_t, t) - \nabla_{\boldsymbol{x}_t} \log p_{0t}(\boldsymbol{x}_t|\boldsymbol{x}_0)\|_2^2 \right] \right\}, \tag{7}$$

where $p_{0t}(\boldsymbol{x}_t|\boldsymbol{x}_0)$ is the transition kernel and can be obtained by the forward process (5). When $\boldsymbol{a}(\cdot, t)$ is affine, the transition kernel is always a Gaussian distribution, where the mean and variance are often known in closed forms (Särkkä and Solin [2019]). With sufficient data and model capacity, score matching ensures that the optimal solution $\boldsymbol{s}_{\boldsymbol{\theta}^*}(\boldsymbol{x}_t, t)$ approximates $\nabla_{\boldsymbol{x}_t} \log p_t(\boldsymbol{x}_t)$ for almost all $\boldsymbol{x}_t$ and $t$.

### 2.2 Gradient guidance

A direct application of diffusion models to CCP (1) is infeasible, as this requires sampling from the product distribution $Q_\beta(\boldsymbol{x}|\rho) \propto r(\boldsymbol{x}|\rho)B_\beta(\boldsymbol{x})$, whereas only samples from $r(\boldsymbol{x}|\rho)$ are accessible. Therefore, obtaining a precise characterization of the score function of $Q_\beta(\boldsymbol{x}|\rho)$ and its diffused version is crucial.

For a given data set $\mathcal{D} = \{(\boldsymbol{x}^{(i)}, \rho^{(i)})\}_{i=1}^{N}$, we use its empirical $p_0(\boldsymbol{x}_0|\rho)$ to approximate the unknown distribution $r(\boldsymbol{x}_0|\rho)$ and denote $\tilde{p}_0(\boldsymbol{x}_0|\rho) \propto p_0(\boldsymbol{x}_0|\rho) B_\beta(\boldsymbol{x}_0)$. The diffused distribution is then given by the forward process (5), i.e.,

$$
\begin{aligned}
p_t(\boldsymbol{x}_t|\rho) &= \int_{\boldsymbol{x}_0} p_{0t}(\boldsymbol{x}_t|\boldsymbol{x}_0) p_0(\boldsymbol{x}_0|\rho) d\boldsymbol{x}_0, \\
\tilde{p}_t(\boldsymbol{x}_t|\rho) &= \int_{\boldsymbol{x}_0} p_{0t}(\boldsymbol{x}_t|\boldsymbol{x}_0) \tilde{p}_0(\boldsymbol{x}_0|\rho) d\boldsymbol{x}_0 \propto \int_{\boldsymbol{x}_0} p_{0t}(\boldsymbol{x}_t|\boldsymbol{x}_0) p_0(\boldsymbol{x}_0|\rho) B_\beta(\boldsymbol{x}_0) d\boldsymbol{x}_0.
\end{aligned}
\tag{8}
$$

In order to sample with the reverse process (6), we need to characterize the score function of the diffused product distribution $\nabla_{\boldsymbol{x}_t} \log \tilde{p}_t(\boldsymbol{x}_t|\rho)$, which is given by the following theorem.

**Theorem 1.** For any given $\beta > 0$, there exists $\hat{\boldsymbol{x}}_0(\boldsymbol{x}_t)$ such that the score function of the diffused product distribution can be formulated as

$$
\nabla_{\boldsymbol{x}_t} \log \tilde{p}_t(\boldsymbol{x}_t|\rho) = \nabla_{\boldsymbol{x}_t} \log p_t(\boldsymbol{x}_t|\rho) \underbrace{-\beta \nabla_{\boldsymbol{x}_t} f(\hat{\boldsymbol{x}}_0(\boldsymbol{x}_t))}_{\text{gradient guidance } \boldsymbol{G}_t},
\tag{9}
$$

where $\nabla_{\boldsymbol{x}_t} \log p_t(\boldsymbol{x}_t|\rho)$ is the score function of the diffused data distribution and $\hat{\boldsymbol{x}}_0(\boldsymbol{x}_t)$ satisfies

$$
f(\hat{\boldsymbol{x}}_0(\boldsymbol{x}_t)) = -\frac{1}{\beta} \log \left( \int_{\boldsymbol{x}_0} p_{t0}(\boldsymbol{x}_0|\boldsymbol{x}_t, \rho) B_\beta(\boldsymbol{x}_0) d\boldsymbol{x}_0 \right).
\tag{10}
$$

Theorem 1 demonstrates that sampling from the product distribution can be accomplished by introducing a gradient guidance term during the sampling process of the original data distribution, which has a strong connection between the posteriori $p_{t0}(\boldsymbol{x}_0|\boldsymbol{x}_t, \rho)$ and the Boltzmann distribution $B_\beta(\boldsymbol{x}_0)$.

Next, we present a special case where the gradient guidance terms admit explicit expressions.

**Corollary 1.** Assume that $p_{t0}(\boldsymbol{x}_0|\boldsymbol{x}_t, \rho) = \mathcal{N}(\boldsymbol{x}_0|\boldsymbol{\mu}_{0|t}, \sigma_{0|t}^2 \boldsymbol{I})$, then we have the following results.

- **First-order guidance:** For $f \in \mathcal{C}^1(\mathbb{R}^n, \mathbb{R})$, we get

$$
\boldsymbol{G}_t = -\beta \nabla_{\boldsymbol{x}_t} f(\boldsymbol{x}_t).
\tag{11}
$$

- **Second-order guidance:** For $f \in \mathcal{C}^2(\mathbb{R}^n, \mathbb{R})$, we get

$$
\boldsymbol{G}_t = -\frac{1}{\sigma_{0|t}^2} \left[ \boldsymbol{H}^{-1} \left( (-\nabla_{\boldsymbol{x}_t}^2 f(\boldsymbol{x}_t) \boldsymbol{x}_t + \nabla_{\boldsymbol{x}_t} f(\boldsymbol{x}_t)) - \frac{1}{\beta \sigma_{0|t}^2} \boldsymbol{\mu}_{0|t} \right) + \boldsymbol{\mu}_{0|t} \right],
\tag{12}
$$

where $\boldsymbol{H} = \nabla_{\boldsymbol{x}_t}^2 f(\boldsymbol{x}_t) + \frac{1}{\beta \sigma_{0|t}^2} \boldsymbol{I}$.

It is worthwhile noting that, for $p_0(\boldsymbol{x}_0|\rho) = \mathcal{N}(\boldsymbol{x}_0|\boldsymbol{\mu}_0, \sigma_0^2 \boldsymbol{I})$ and the Gaussian transition kernel, the assumption in Corollary 1 holds and the parameters $(\boldsymbol{\mu}_{0|t}, \sigma_{0|t})$ can be expressed explicitly. In practice, we can use Tweedie's formula (Efron [2011]) to obtain an estimator of $\boldsymbol{\mu}_{0|t}$, and treat the variance as a hyper parameter; see Subsection 3.3 for details on this. Although the second-order guidance requires computing the inverse of a general Hessian matrix, which may be computationally expensive, it brings faster convergence and better variance reduction.

## 3  GGDOpt for CCP

In this section, we give our GGDOpt framework for CCP (1). The whole process can be divided into three stages: data generation, diffusion and learning, and sampling with guidance. More specifically, in the data generation stage, a collection of points satisfying the chance constraint is generated to characterize the nonconvex feasible set. The diffusion and learning stage progressively inject noise to convexify the nonconvex feasible region and learn the score function of the conditional distribution in order to perform sampling. After learning, the sampling with guidance stage iteratively runs the reverse process with an extra gradient guidance to sample from the product distribution, which will asymptotically converge to an optimal solution to problem (1). Next, we present the details of the three stages in GGDOpt one by one.

## 3.1 Stage 1: data generation

First we give an efficient approach to generate high-quality data that satisfy the chance constraint while maintaining lower objective values. Suppose that we have a set of samples $\{h^{(\ell)}\}_{\ell=1}^L$, denote the empirical mean $\bar{h} = \frac{1}{L} \sum_{\ell=1}^L h^{(\ell)}$. Notice that in most of cases, it's much easier to solve the following deterministic restricted problem (RP) with a fixed $\bar{h}$:

$$\min_{\boldsymbol{x}} \quad f(\boldsymbol{x})$$
$$\text{s.t.} \quad \boldsymbol{g}(\boldsymbol{x}, \bar{\boldsymbol{h}}) \geq \boldsymbol{z}_i, \tag{13}$$

where $\boldsymbol{z}_i \geq \boldsymbol{0}$ is a given restriction, $i = 1, \ldots, N$. Let $\boldsymbol{x}(\boldsymbol{z}_i)$ denote the solution to problem (13) for a given $\boldsymbol{z}_i$. As the smallest element $z_{min}$ in $\boldsymbol{z}_i$ increases, the probability of the nonlinear constraint $g(\boldsymbol{x}(\boldsymbol{z}_i), \boldsymbol{h}) \geq 0$ also increases. Then, solving problem (13) allows us to generate high-quality data that satisfies the chance constraint for arbitrary $\rho \in (0, 1)$ while enjoys low objective values.

Since the distribution of the random variable $\boldsymbol{h}$ is unknown, referring SAA method, we approximate the chance constraint using the empirical distribution over samples $\{h^{(\ell)}\}_{\ell=1}^L$. Then, after getting $\boldsymbol{x}(\boldsymbol{z}_i)$, we have

$$\text{Prob}_{\boldsymbol{h}}\{\boldsymbol{g}(\boldsymbol{x}(\boldsymbol{z}_i), \boldsymbol{h}) \geq \boldsymbol{0}\} \approx \underbrace{\frac{1}{L} \sum_{l=1}^L \ell_{0/1}(\boldsymbol{g}(\boldsymbol{x}(\boldsymbol{z}_i), \boldsymbol{h}^{(\ell)}))}_{1-\rho^{(i)}}, \tag{14}$$

where $\ell_{0/1}(\boldsymbol{g}) = 1$ if $\boldsymbol{g} \geq \boldsymbol{0}$ and $\ell_{0/1}(\boldsymbol{g}) = 0$ otherwise. By calculating the empirical $\rho^{(i)}$, an asymptotic approximation of the underlying probability is obtained, requiring no assumption on the underlying distribution $P$. In the appendix, we give a tight lower bound for the probability constraint $\text{Prob}_{\boldsymbol{h}}\{\boldsymbol{g}(\boldsymbol{x}(\boldsymbol{z}_i), \boldsymbol{h}) \geq \boldsymbol{0}\}$ if the variance and the mean of the random variable $\boldsymbol{h}$ are known, which is helpful to obtain a better approximation $\rho^{(i)}$.

Let $\boldsymbol{x}^{(i)} := \boldsymbol{x}(\boldsymbol{z}_i)$ and repeating the above process, i.e., solving problem (13) and estimating $\rho^{(i)}$, and gradually increasing $\boldsymbol{z}_i$, we can generate a collection of data points $\mathcal{D} = \{\boldsymbol{x}^{(i)}, \rho^{(i)}\}_{i=1}^N$, which are then used to train our GGDOpt in the next stages.

## 3.2 Stage 2: diffusion and learning

From Theorem 1, we observe that the score function of the diffused product distribution has two terms, the conditional score $\nabla_{\boldsymbol{x}_t} \log p_t(\boldsymbol{x}_t|\rho)$ and the gradient guidance term $\boldsymbol{G}_t$ for which explicit forms of first- and second-order guidances have been derived in Corollary 1. Then the challenge reduces to learning the conditional score $\nabla_{\boldsymbol{x}_t} \log p_t(\boldsymbol{x}_t|\rho)$.

In practice, naively conditioning a standard diffusion model by appending the conditioned variable at each step of the sampling process does not work well, as the model often ignores the conditioned information. Related works on conditional score estimation have been studied in (Dhariwal and Nichol [2021], Dhariwal and Nichol [2021], Ho and Salimans [2022]). Here we propose to use the classifier-free guidance (Ho and Salimans [2022]) to give an approximation of $\nabla_{\boldsymbol{x}} \log p_t(\boldsymbol{x}|\rho)$.

Instead of training a separate classifier model, classifier-free guidance choose to train an unconditional score estimator to approximate $\nabla_{\boldsymbol{x}_t} \log p_t(\boldsymbol{x}_t)$ together with the conditional score estimator to approximate $\nabla_{\boldsymbol{x}_t} \log p_t(\boldsymbol{x}_t|\rho)$. Specificity, we train a single model $\boldsymbol{s}_{\boldsymbol{\theta}}(\boldsymbol{x}_t, t, \rho)$, and the conditioning information $\rho$ is randomly discarded as empty set $\emptyset$ with probability $p_{uncond}$ to train unconditionally. Then the conditional score $\nabla_{\boldsymbol{x}_t} \log p_t(\boldsymbol{x}_t|\rho)$ is estimated by

$$\nabla_{\boldsymbol{x}_t} \log p_t(\boldsymbol{x}_t|\rho) \approx (1+w)\boldsymbol{s}_{\boldsymbol{\theta}}(\boldsymbol{x}_t, t, \rho) - w\boldsymbol{s}_{\boldsymbol{\theta}}(\boldsymbol{x}_t, t, \emptyset), \tag{15}$$

for a given weight parameter $w$. Specifically, for the given data set $\mathcal{D}$ and network $\boldsymbol{s}_{\boldsymbol{\theta}}(\boldsymbol{x}_t, t, \rho)$ parameterized by $\boldsymbol{\theta}$, the training objective is defined as

$$\text{Loss}(\boldsymbol{\theta}) = \mathbb{E}_{t \sim \mathcal{U}[0,T]} \left\{ \mathbb{E}_{\boldsymbol{x}_0, \rho} \mathbb{E}_{\boldsymbol{x}_t|\boldsymbol{x}_0} \left[ \|\boldsymbol{s}_{\boldsymbol{\theta}}(\boldsymbol{x}_t, t, \rho) - \nabla_{\boldsymbol{x}_t} \log p_{0t}(\boldsymbol{x}_t|\boldsymbol{x}_0)\|_2^2 \right] \right\}, \tag{16}$$

and trained with Adam (Kingma [2014]). The training process of GGDOpt is given in Algorithm 1.

| **Algorithm 1** Training of GGDOpt | **Algorithm 2** Sampling of GGDOpt |
|---|---|
| **Input:** $\{(\boldsymbol{x}^{(i)}, \rho^{(i)})\}_{i=1}^{N} \sim p_0(\boldsymbol{x}\|\rho)$. | **Input:** $\boldsymbol{s}_{\boldsymbol{\theta}^*}(\boldsymbol{x}, t, \rho)$, objective $f$. |
| **Output:** $\boldsymbol{s}_{\boldsymbol{\theta}^*}(\boldsymbol{x}, t, \rho)$. | **Output:** $\boldsymbol{x}_0^*$. |
| 1: **repeat** | 1: $\boldsymbol{x}_T \sim p_T$. |
| 2:    Load $(\boldsymbol{x}_0, \rho_0) \sim p_0(\boldsymbol{x}\|\rho)$. | 2: **for** $t = T, ..., 1$ **do** |
| 3:    Set $\rho \leftarrow \emptyset$ with probability $p_{uncond}$. | 3:    Calculate $\tilde{\boldsymbol{s}}_{\boldsymbol{\theta}}(\boldsymbol{x}_t, t, \rho)$ with (18). |
| 4:    Sample $t \sim \mathcal{U}[0, T]$. | 4:    Calculate $\boldsymbol{G}_t$ with (11) or (12) . |
| 5:    Generate $\boldsymbol{x}_t \sim p_{0t}(\boldsymbol{x}_t\|\boldsymbol{x}_0)$. | 5:    Take guided sampling step with (17). |
| 6:    Take gradient descent step on (16). | 6: **end for** |
| 7: **until** converged. | 7: **return** $\boldsymbol{x}_0^* = \boldsymbol{x}_0$. |

### 3.3 Stage 3: sampling with guidance

Given the forward process (5), the corresponding reverse process is given by the following reverse-time SDE with trained $\boldsymbol{s}_{\boldsymbol{\theta}}(\boldsymbol{x}_t, t, \rho)$ and gradient guidance $\boldsymbol{G}_t$:

$$d\boldsymbol{x}_t = \left[\boldsymbol{a}(\boldsymbol{x}_t, t) - b(t)^2\big(\tilde{\boldsymbol{s}}_{\boldsymbol{\theta}}(\boldsymbol{x}_t, t, \rho) + \boldsymbol{G}_t\big)\right] dt + b(t)d\bar{\boldsymbol{B}}_t, \tag{17}$$

where

$$\tilde{\boldsymbol{s}}_{\boldsymbol{\theta}}(\boldsymbol{x}_t, t, \rho) = (1 + w)\boldsymbol{s}_{\boldsymbol{\theta}}(\boldsymbol{x}_t, t, \rho) - w\boldsymbol{s}_{\boldsymbol{\theta}}(\boldsymbol{x}_t, t, \emptyset). \tag{18}$$

For the first-order gradient guidance $\boldsymbol{G}_t$ in (11), we directly use the gradient of the objective scaled by a hyper parameter $\beta$. For the second-order gradient guidance (12), we need to give the posterior mean and variance $(\boldsymbol{\mu}_{0|t}, \sigma_{0|t}^2)$. Here we use Tweedie's formula (Efron [2011]) to get an estimator of the posterior mean as follows:

$$\boldsymbol{\mu}_{0|t} = \mathbb{E}\left[\boldsymbol{x}_0|\boldsymbol{x}_t, \rho\right] = \frac{1}{\sqrt{\bar{\alpha}_t}}(\boldsymbol{x}_t + (1 - \bar{\alpha}_t)\tilde{\boldsymbol{s}}_{\boldsymbol{\theta}}(\boldsymbol{x}_t, t, \rho)), \tag{19}$$

with priori $p_{0t}(\boldsymbol{x}_t|\boldsymbol{x}_0) = \mathcal{N}(\boldsymbol{x}_t|\sqrt{\bar{\alpha}_t}\boldsymbol{x}_0, (1 - \bar{\alpha}_t)\boldsymbol{I})$ for a specific noising schedule $\bar{\alpha}_t$.

While Tweedie's formula theoretically provides both the posterior mean and covariance, $\boldsymbol{\Sigma}_{0|t} = (1 - \bar{\alpha}_t)(\boldsymbol{I} + (1 - \bar{\alpha}_t)\nabla^2 \log p(\boldsymbol{x}_t))$, computing the covariance requires evaluating the Hessian of $\log p(\boldsymbol{x})$. In our framework, the score function $\boldsymbol{s}_{\boldsymbol{\theta}}$ is parameterized by a neural network, and computing its second derivatives involves backpropagation through the network's Jacobian, which is computationally expensive, especially in high dimensions. To strike a balance between performance and efficiency, we choose to treat the covariance as a tunable hyper parameter $\sigma^2$. In the appendix, we give a detailed comparison between the fully Tweedie-based method and our approach to show that using a fixed variance can be a practical and robust alternative.

Then the second-order guidance can be calculated by

$$\boldsymbol{G}_t = -\frac{1}{\sigma^2}\left[(\nabla^2 f(\boldsymbol{x}_t) + \frac{1}{\beta\sigma^2}\boldsymbol{I})^{-1}\left((-\nabla^2 f(\boldsymbol{x}_t)\boldsymbol{x}_t + \nabla f(\boldsymbol{x}_t)) - \frac{1}{\beta\sigma^2}\boldsymbol{\mu}_{0|t}\right) + \boldsymbol{\mu}_{0|t}\right], \quad (20)$$

and the sampling process of GGDOpt is given in Algorithm 2.

## 4 Convergence analysis

In this section, we give the convergence analysis of the proposed GGDOpt framework in both theoretical and practical aspects. We show that: theoretically, the samples generated by the sampling process will concentrate around the points with the lowest function values within the support of the data distribution; and practically, the gap between the expected function values of generated samples and the optimal value will be bounded by two components.

### 4.1 Theoretical convergence

As provided by (Pidstrigach [2022]), under mild assumptions, the sampling distribution of the standard diffusion model will have the exact same support as the data distribution. But what if we introduce an

extra gradient guidance term? For a given $\rho$, denote $\mathcal{D}_\rho = \{\boldsymbol{x}^{(i)} \mid (\boldsymbol{x}^{(i)}, \rho^{(i)}) \in \mathcal{D}, \rho^{(i)} \leq \rho\}$ as the approximated feasible set of $\mathcal{X}_\rho$. The following theorem says that in our settings, as $T \to \infty$ and $\beta \to \infty$, the samples of GGDOpt will concentrate around the points with the lowest function values within the support of the data distribution $\mathcal{D}_\rho$ for any given $\rho$.

**Theorem 2.** For any given $\rho \in (0, 1)$, suppose that there exists a constant $\delta$ such that the error in the score estimation can be bounded as:
$$\|\tilde{\boldsymbol{s}}_{\boldsymbol{\theta}}(\boldsymbol{x}_t, t, \rho) + \boldsymbol{G}_t - \nabla_{\boldsymbol{x}_t} \log \tilde{p}_t(\boldsymbol{x}_t|\rho)\| \leq \delta, \quad \forall \, \boldsymbol{x}_t. \tag{21}$$
For samples $\tilde{\boldsymbol{x}}_{sample} \sim p_{sample}(\boldsymbol{x}_0|\rho)$ generated by the reverse process
$$d\boldsymbol{x}_t = \big[\boldsymbol{a}(\boldsymbol{x}_t, t) - b(t)^2\big(\tilde{\boldsymbol{s}}_{\boldsymbol{\theta}}(\boldsymbol{x}_t, t, \rho) + \boldsymbol{G}_t\big)\big] dt + b(t)d\bar{\boldsymbol{B}}_t, \tag{22}$$
with prior $p_{prior} = \mathcal{N}(\boldsymbol{0}, \boldsymbol{I})$, affine drift coefficients $\boldsymbol{a}(\cdot, t)$, and
$$\tilde{\boldsymbol{s}}_{\boldsymbol{\theta}}(\boldsymbol{x}_t, t, \rho) = (1 + w)\boldsymbol{s}_{\boldsymbol{\theta}}(\boldsymbol{x}_t, t, \rho) - w\boldsymbol{s}_{\boldsymbol{\theta}}(\boldsymbol{x}_t, t, \emptyset), \tag{23}$$
as $T \to \infty$, $p_{sample}(\boldsymbol{x}_0|\rho)$ will have the same support as $\tilde{p}_0(\boldsymbol{x}_0|\rho)$. Further, as $\beta \to \infty$, $\tilde{\boldsymbol{x}}_{sample}$ will concentrate around $\boldsymbol{x}^* = \arg\min_{\boldsymbol{x} \in \mathcal{D}_\rho} f(\boldsymbol{x})$.

The assumption in the score estimation error (21) quantifies the approximation accuracy of the trained score network relative to the true score function. It depends on the training quality of the neural network and the expressiveness of the model class. This type of assumption is common in the theoretical analysis of diffusion models (see, e.g., Pidstrigach [2022], De Bortoli et al. [2021]) and is used to establish convergence results in generative modeling and sampling.

## 4.2 Practical error bound

In practice, the forward process cannot reach the stationary distribution and the training is not perfect. This results in the failure of the sample distribution to strictly concentrate on the data points. This will lead to two components of errors: the limited time length error $I_1$ and limited inverse temperature error $I_2$, which are given as follows:
$$|\mathbb{E}[f(\tilde{\boldsymbol{x}}_{sample})] - f(\boldsymbol{x}^*)| \leq \underbrace{|\mathbb{E}[f(\tilde{\boldsymbol{x}}_{sample})] - \mathbb{E}[f(\boldsymbol{x}^\pi)]|}_{I_1} + \underbrace{|\mathbb{E}[f(\boldsymbol{x}^\pi)] - f(\boldsymbol{x}^*)|}_{I_2}. \tag{24}$$
In the above, $\tilde{\boldsymbol{x}}_{sample}$ is sampled from the reverse process (17), $\boldsymbol{x}^\pi$ follows the strong solution $p^\pi$ to the Fokker-Planck equation of (17), and $\boldsymbol{x}^* = \arg\min_{\boldsymbol{x} \in \mathcal{D}_\rho} f(\boldsymbol{x})$. Next, we will give practical error bounds of both the two components with finite $T$ and $\beta$.

**Assumption 1.** We assume the following conditions hold:

- The forward process is given by $d\boldsymbol{x} = b(t)d\boldsymbol{B}_t$;
- The reverse process starts in $p_{prior} = \mathcal{N}(\boldsymbol{m}_T, \boldsymbol{\Sigma}_T)$ where $\boldsymbol{m}_T = \mathbb{E}[\tilde{p}_0(\boldsymbol{x}_0|\rho)]$ and $\boldsymbol{\Sigma}_T = \mathrm{Cov}(\tilde{p}_0(\boldsymbol{x}_0|\rho)) + T \cdot \boldsymbol{I}$;
- The objective function $f(\boldsymbol{x})$ satisfies $\|\nabla_{\boldsymbol{x}} f(\boldsymbol{x})\|_2 \leq C_1 \|\boldsymbol{x}\|_2 + C_2$.

The first two conditions in Assumption 1 correspond to the VE SDE in (Song et al. [2020b]) and are primarily used to characterize the discrepancy between the end distribution and the prior distribution. The third assumption is common in the convergence analysis of stochastic optimization and sampling algorithms (see, e.g., Raginsky et al. [2017]). In practice, Assumption 1 holds for a broad class of functions, including smooth bounded functions and quadratic objectives, which frequently arise in real-world optimization problems.

**Theorem 3.** Under Assumption 1, denote $\sigma^{(k)}, k = 1, \ldots, n$, the eigenvalues of $\boldsymbol{\Sigma}_T$. For any given $\rho \in (0, 1)$, denote $N_\rho = |\mathcal{D}_\rho|$ and $\boldsymbol{x}^* = \arg\min_{\boldsymbol{x} \in \mathcal{D}_\rho} f(\boldsymbol{x})$. Then for any given $T > 0$ and $\beta > 0$, the optimization error can be bounded by
$$|\mathbb{E}[f(\tilde{\boldsymbol{x}}_{sample})] - f(\boldsymbol{x}^*)| \leq \underbrace{C_I\big(\sqrt{C_T} + (C_T/2)^{1/4}\big)}_{I_1} + \underbrace{(N_\rho - 1) \max_{\boldsymbol{x} \in \mathcal{D}_\rho} |f(\boldsymbol{x}) - f(\boldsymbol{x}^*)|e^{-\beta\delta_\rho}}_{I_2},$$
$$\tag{25}$$
where $C_T = \frac{1}{2}\log\big(\prod_{k=1}^n (\sigma^{(k)}/T)\big)$ and $C_I, \delta_\rho$ are constants.

Theorem 3 provides a non-asymptotic convergence result of GGDOpt with limited time length and inverse temperature. As $T \to \infty$ and $\beta \to \infty$, the optimization error goes to zero and GGDOpt is shown to generate asymptotically optimal solutions.

# 5 Experimental results

In this section, we perform numerical experiments on both synthetic datasets and a wireless communications waveform design problem. To generate the data, we employ CVX (Grant et al. [2008]) to solve the restricted problem (13). In the diffusion and learning stage, we set $T = 1000$ with a linear noise schedule $\eta(t)$ ranging from 0.0001 to 0.02, and let $\boldsymbol{a}(\boldsymbol{x}, t) = -\frac{1}{2}\eta(t)\boldsymbol{x}$ and $b(t) = \sqrt{\eta(t)}$. In the sampling with guidance stage, we evaluate both first- and second-order gradient guidances via implementing a DDIM-based technique (Song et al. [2020a]) with a descaled time step $T' = 100$ for accelerated sampling. We employ two variants of the U-Net model (Ronneberger et al. [2015]) as our score estimator: U-Net-1D for the linear chance constrained problem and both for robust waveform design. Additional experimental details are provided in the supplementary materials.

## 5.1 Linear chance constrained problem

Consider the following linear chance constrained problem:

$$
\begin{aligned}
\min_{\boldsymbol{x} \in \mathbb{R}^n} \quad & \frac{1}{2}\boldsymbol{x}^\top \boldsymbol{x} + \boldsymbol{b}^\top \boldsymbol{x} \\
\text{s.t.} \quad & \mathrm{Prob}_{\boldsymbol{c} \sim p_{\boldsymbol{c}}}\{\boldsymbol{c}^\top \boldsymbol{x} + d \geq 0\} \geq 1 - \rho,
\end{aligned}
\tag{26}
$$

where $p_{\boldsymbol{c}} = \mathcal{N}(\boldsymbol{c}; \bar{\boldsymbol{c}}, \boldsymbol{I})$ and $(\boldsymbol{b}, \bar{\boldsymbol{c}}, d, \rho)$ are hyper parameters selected from a test set. The above problem can be reformulated as a second-order conic (SOC) program, for which CVX (Grant et al. [2008]) is used for solution. To generate training data, we solve the restricted version of problem (26) for $N = 1000$ values of $z$ linearly spaced in the interval $[0, 0.5]$. Then we execute the reverse process with first- and second-order gradient guidance to generate samples.

We compare our proposed GGDOpt against different types of SAA methods for solving the problem, using the corresponding CVX solutions as performance benchmarks. Each algorithm was executed 100 times (except CVX). The results with $n = 8$ are presented in Table 1.

Table 1: Comparison results on the linear chance constrained problem (26)

| Method | Repeat | FvalMean | FvalStd | FvalMedian | Runtime |
|---|---|---|---|---|---|
| SOC_CVX (Grant et al. [2008]) | 1 | **-0.6586** | 0 | -0.6586 | 0.3214 |
| SAA_CVaR (Nemirovski and Shapiro [2007]) | 100 | -0.5893 | 0.0248 | -0.5869 | 0.3063 |
| SAA_MIP (Pagnoncelli et al. [2009]) | 100 | -0.6281 | 0.0157 | -0.6318 | 15.4502 |
| SAA_PDCA (Wang et al. [2023]) | 100 | -0.6389 | 0.0314 | -0.6408 | 0.6276 |
| SAA_SNSCO (Zhou et al. [2024]) | 100 | 0.8051 | 3.4014 | -0.6371 | 0.2793 |
| GGDOpt_WithoutGuidance | 100 | 0.3481 | 0.5486 | 0.2798 | 0.0465 |
| GGDOpt_First-order | 100 | -0.6483 | **0.0051** | -0.6488 | **0.0486** |
| GGDOpt_Second-order | 100 | **-0.6491** | **0.0056** | **-0.6503** | **0.0507** |

The results in Table 1 demonstrate that, compared to the SOC_CVX method, which requires explicit knowledge of the underlying distribution, GGDOpt can approximately find the global minimizer with only samples from distribution $p_{\boldsymbol{c}}$ while simultaneously achieving significant overhead reduction. Compared to SAA methods, GGDOpt achieves superior performance in terms of lower function values and enhanced numerical stability under the effect of gradient guidance.

As expected, the runtime increases with the problem dimension. However, both the first- and second-order versions of GGDOpt remain consistently faster than the baseline SAA_PDCA method across all dimensions. Moreover, the increase in runtime is moderate, indicating that our approach scales favorably even in high-dimensional settings.

Furthermore, as the runtime increases with the problem dimension, both the first- and second-order versions of GGDOpt reduce the computational time by approximately 80% compared with , offering substantial efficiency improvements. More detailed experimental results on larger problem scale and computational costs are listed in the appendix.

## 5.2 Robust waveform design

Consider the following robust waveform design problem (Wang et al. [2014])

$$\min_{\boldsymbol{S}_1,\ldots,\boldsymbol{S}_K \in \mathbb{R}^{N_t \times N_t}} \sum_{i=1}^{K} \mathrm{Tr}(\boldsymbol{S}_i)$$
$$\text{s.t.} \quad \mathrm{Prob}_{\boldsymbol{h}_i \sim \mathcal{N}(\bar{\boldsymbol{h}}_i, \boldsymbol{C}_i)}\{\mathrm{R}_i \geq r_i\} \geq 1 - \rho_i, i = 1, 2, \ldots, K, \quad (27)$$
$$\boldsymbol{S}_1, \ldots, \boldsymbol{S}_K \succeq \boldsymbol{0}, i = 1, 2, \ldots, K,$$

where $N_t$ is the number of antennas at the base station and $K$ is the total number of users. For each user $i$, $\boldsymbol{S}_i \succeq \boldsymbol{0}, \boldsymbol{h}_i, R_i$ and $r_i \geq 0$ denote the signal covariance matrix (to be designed), the random channel vector, the achievable rate, and the desired rate target, respectively.

Firstly, we use U-Net-2D as the score estimator. Notice that during the data generation, all the solutions to the restricted problem (13) exhibit a rank-one structure (Huang and Zhang [2007], Chang et al. [2008], Huang et al. [2020]). Remarkably, the generated samples maintain this rank-one property (with dominant eigenvalue accounting for >99% of the total eigenvalue) after training, suggesting that the solutions to the robust waveform design problem (27) inherently reside on a rank-one manifold with extremely high probability (Wang et al. [2014]), which GGDOpt successfully captures. This implies that rank-one decomposition can be effectively applied after generation, enabling the use of U-Net-1D as a score estimator to reduce computational costs in both training and sampling process.

Table 2 summarizes the comparison results of GGDOpt and two state-of-the-art methods for solving problem (27) with $N_t = 16$ and $K = 3$, where the worst probabilities that the chance constraints satisfy for $K$ users are underlined. Notably, both baseline methods rely on explicit knowledge of the underlying distribution, whereas GGDOpt operates solely based on samples. The results show that GGDOpt outperforms existing convex approximation methods, achieving superior feasible solutions outside the convex restriction of the feasible set, while significantly reducing computational overhead. Complete experimental details are provided in the appendix.

Table 2: Optimization methods comparison for robust waveform design

| Method | Metric | $\rho = 0.05$ | $\rho = 0.10$ | $\rho = 0.15$ | $\rho = 0.20$ |
|---|---|---|---|---|---|
| Sphere Bounding Ben-Tal and Nemirovski [2000] | Probability | 0.99; 0.99; 0.99 | 0.99; 0.99; 0.99 | 0.99; 0.99; 0.99 | 0.99; 0.99; 0.99 |
| | FuncValue | 0.1374 | 0.1366 | 0.1361 | 0.1357 |
| | Runtime | 1.4688 | 1.4375 | 1.4113 | 1.3875 |
| Bernstein-type Inequality Wang et al. [2014] | Probability | 0.96; 0.95; 0.96 | 0.93; 0.93; 0.93 | 0.91; 0.91; 0.92 | 0.90; 0.90; 0.91 |
| | FuncValue | 0.1260 | 0.1253 | 0.1248 | 0.1244 |
| | Runtime | 1.2938 | 1.2813 | 1.2593 | 1.2652 |
| GGDOpt First-order guidance | Probability | 0.99; 0.95; 0.99 | 0.92; 0.98; 0.91 | 0.93; 0.86; 0.94 | 0.87; 0.81; 0.91 |
| | FuncValue | 0.1279 | 0.1265 | 0.1254 | 0.1247 |
| | Runtime | 0.0691 | 0.0628 | 0.0603 | 0.0635 |
| GGDOpt Second-order guidance | Probability | 0.97; 0.95; 0.96 | 0.90; 0.94; 0.90 | 0.88; 0.85; 0.86 | 0.88; 0.80; 0.87 |
| | FuncValue | 0.1260 | **0.1246** | **0.1239** | **0.1237** |
| | Runtime | **0.0788** | **0.0712** | **0.0687** | **0.0682** |

## 6 Conclusion

In this paper, we have proposed GGDOpt, a gradient-guided diffusion framework that efficiently solves nonconvex CCP without requiring the exact distribution knowledge. By reformulating CCP as a sampling problem over the product of an unknown data distribution and a Boltzmann distribution, GGDOpt leverages both first- and second-order gradient information during reverse sampling. Theoretical convergence guarantees and practical error bounds are provided under mild assumptions. Experimental results demonstrate that GGDOpt outperforms existing methods in both solution quality and numerical stability with significant overhead reduction.

## Acknowledgments

The work of Boyang Zhang and Ya-Feng Liu was supported in part by the National Natural Science Foundation of China (NSFC) under Grant 12021001 and Grant 12371314. The work of Zhiguo Wang was supported in part by The National Key Research and Development Program of China under Grant 2020YFA0714003 and in part by NSFC under Grant 62203313.

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
