# OpenReview forum: "A Gradient Guided Diffusion Framework for Chance Constrained Programming"
_NeurIPS.cc/2025/Conference — NeurIPS 2025 poster_

### Official Review · Reviewer_u5CA · 2025-06-30

**Clarity:** 3
**Significance:** 2
**Originality:** 2
**Rating:** 4
**Confidence:** 3

**Summary:**

The paper develops a gradient-guided diffusion framework to solve chance-constrained programming problems without knowing the exact distribution. The framework first samples solutions that satisfy the chance constraints approximately. Next, the paper employs classifier-free diffusion model training and model-guided sampling to build the diffusion model. The model guidance is derived through a Boltzmann distribution defined by the objective function. The proposed algorithm is evaluated for a linear chance constrained problem and a robust waveform design. Overall, the paper applies the classifier-free and model-guided diffusion-based method for a traditional optimization problem: $\max f(x)$, subject to a chance constraint $\mathbb{P}_h(g(x,h)>0)\geq 1-\rho$. The method is kind of standard, but it is first applied in the chance-constrained problem.

**Questions:**

See Strengths And Weaknesses.

**Ethical Concerns:**

["NO or VERY MINOR ethics concerns only"]

**Final Justification:**

The authors provided additional experimental results and revised some theoretical analysis during the rebuttal process. Although there are some technical flaws in the current version, the authors give a plan to address them. Given that this paper has a potential to become a rigorous and complete one, I remain a borderline rating but lean toward acceptance.

**Limitations:**

Yes.

**Quality:**

2

**Strengths And Weaknesses:**

The paper has the following strengths:
+ The paper proposes a reformulation for chance-constrained optimization problems in Eqn. (4) which utilizes Boltzmann distribution and indicator function to transform the chance-constrained optimization to an unconstrained optimization problem.
+ Assuming that the backward process starts from a Gaussian distribution and the gradient of the objective function is bounded, a convergence analysis and regret analysis are provided to theoretically understand the performance. The expected error between the output and the optimal solution is bounded by some proof techniques.
+ The paper evaluates the methods on different chance-constrained optimization with real applications and compares the proposed method with several baselines.  The applications include linear chance constrained problem and robust waveform design. The baselines include CVX and SAA methods. They show some performance improvement but it is uncertain whether the baselines are compared in the same level of chance constraint violation.
+ The paper is well-written.


The paper has the following weaknesses:
- The method cannot guarantee the satisfaction of the chance constraints since the exact distribution is unknown. The authors didn't discuss the constraint violation either theoretically or experimentally. The authors did not show the violation of the constraints in the experiments in Section 5. I'm concerned that the comparison in the experiments may not be fair since the violations of different algorithms can be different.
- It is unclear why the proposed method is better than traditional optimization methods. Also, why is the diffusion-based method better than learning-to-optimize algorithms with classical neural networks? Why not use other generative models like autoregressive models? The discussions on the motivation of using diffusion models in this problem is not enough.
- The paper employs the existing methods of the classifier-free diffusion model and gradient-guided sampling on the problem of chance-constrained optimization. The algorithm lacks enough originality.  The algorithms in Algorithm 1 and Algorithm 2 are standard classifier-free and gradient-guided conditional diffusion-based methods.  There is limited novelty in terms of the methods on conditional diffusion models.
- The complexity of training a diffusion model and sampling from a diffusion model is usually very large. It would be better to have a complexity analysis for the proposed algorithm.

---

> ### Author Rebuttal · Authors · 2025-07-31
>
> **Reply**
>
> Many thanks for your detailed and insightful comments, which help us clarify important points and improve our manuscript. We hope the following point-by-point responses can address your concerns and offer a clearer understanding of our work.
>
> **Reply to Weakness 1:**
>
> Thank you for raising this important point regarding the constraint violation. We provide a comprehensive discussion from both theoretical and empirical perspectives to clarify how our method handles violations of the chance constraints.
>
> -**Theoretical Guarantee:**
> Given a well-trained diffusion model whose score network closely approximates the true score, Theorem 2 guarantees that the samples generated by GGDOpt converge to the optimal solution of the chance constrained program. In particular, in the limit of infinite sampling time and model capacity, the generated solutions satisfy the chance constraints with high probability.
>
> -**Experimental Verification:**
> In Section 5.1, we focus on linear chance constrained problems under Gaussian distribution, where the chance constraint can be equivalently reformulated as a second-order cone (SOC) constraint. To ensure a fair comparison across all methods, we project the final solutions generated by all algorithms, including GGDOpt and some baseline methods, onto the feasible SOC region. This ensures that all reported solutions satisfy the chance constraints.
>
> In Section 5.2, the analytical form of the chance constraint is not available. Therefore, we estimate the constraint satisfaction empirically. For each solution, we draw a large number of samples from the uncertainty distribution and compute the empirical probability. These results are reported in the table shown below, which includes multiple uncertainty levels. For example, when $\rho=0.05$, a solution is considered feasible if its estimated probability of satisfaction exceeds 0.95.
>
> **Table:Empirical feasibility of different algorithms under some uncertainty levels.**
> |Method|$\rho=0.05$|$\rho=0.10$|$\rho=0.15$|$\rho=0.20$|
> |---|---|---|---|---|
> |Sphere Bounding|0.99|0.99|0.99|0.99|
> |Bernstein-type Inequality|0.96|0.93|0.91|0.90|
> |GGDOpt(First-Order)|0.99|0.92|0.93|0.87|
> |GGDOpt(Second-Order)|0.97|0.90|0.88|0.88|
>
> As expected, classical convex approximation methods produce highly conservative solutions that strictly satisfy the chance constraints. GGDOpt, on the other hand, generates solutions closer to the boundary of the feasible region. Nevertheless, GGDOpt consistently achieve estimated satisfaction probabilities near the required threshold, demonstrating practical feasibility.
>
> **Reply to Weakness 2:**
>
> Thank you for this important comment, which gives us an opportunity to clarify the motivation and contributions of the proposed GGDOpt framework.
>
> -**Unknown distribution and computational efficiency.**
> There are two traditional optimization methods, which are Convex Approximation (CA) and Sample Average Approximation (SAA) methods. The CA method seeks to construct an inner approximation of the chance constraint, but it typically requires the exact distribution. The SAA method reformulates CCP as a binary integer program, which remains computationally inefficient. Unlike classical convex approximation approaches for CCP, our method does **not require prior knowledge of the underlying distribution**, which makes our approach applicable to broader and more realistic settings where the distribution is unknown.
>
> -**Sampling can be faster.**
> Recent studies (e.g., [Ma2019]) show that sampling-based methods can outperform optimization-based methods in nonconvex landscapes. The chance constraints are often nonconvex and do not admit a closed-form formulation, which poses major challenges for direct optimization. While Learning-to-Optimize (L2O) is a promising line of work, we are not aware of any L2O algorithm specifically designed for general CCPs. A key reason is that L2O methods often require a strong base optimizer for supervision or unfolding. The chance constraints have no explicit analytical form, making them incompatible with existing L2O pipelines.
>
> -**Theoretical convergence.**
> Compared to other generative models such as VAEs and GANs, diffusion models offer stronger theoretical support for convergence, especially in high-dimensional and multimodal scenarios. We reformulate the original CCP as a sampling problem over the product of two distributions. This novel perspective allows us to cast CCP as a generative problem and leverage the reverse-time SDE underlying diffusion models. Under this formulation, we prove that the samples generated by GGDOpt converge to the optimal solution.
>
> **Reply to Weakness 3:**
>
> The rising prominence of diffusion models has spurred significant research interest in their underlying mathematical foundations and applications. While guidance techniques have been widely applied in areas such as image generation, black-box optimization, and inverse problems, our contribution does not lie in introducing guidance, but rather in deriving **a novel class of guidance terms based on product distributions**, specifically tailored to chance constrained problems with unknown distributions.
>
> More specifically, compared to prior work such as [Guo2024], our approach introduces two main innovations:
>
> - **Conditional Training and Applicability Beyond Linear-Gaussian Settings:**  Unlike [Guo2024], which applies guidance to pre-trained unconditional diffusion models and assumes a linear objective with Gaussian data, our framework involves a dedicated data generation process followed by conditional score training. This enables us to address nonlinear and structurally complex chance constrained problems, where directly sampling from the feasible region is nontrivial.
>
> - **A New Class of Guidance Derived from Product Distributions:** In our work, we derive two types of guidance terms directly from the product distribution formulation of the target density:
>
> 1. a first-order guidance
>
> $$\boldsymbol{G}_t^{(1)} = - \beta \nabla f(\boldsymbol{x}_t),$$
>
> 2. a second-order guidance
>
> $$\boldsymbol{G}_t^{(2)} = -\frac{1}{\sigma^2}[\boldsymbol{H}^{-1}[(\nabla^2 f(\boldsymbol{x}_t) \boldsymbol{x}_t+\nabla f(\boldsymbol{x}_t)) -\frac{1}{\beta \sigma^2}\boldsymbol{\mu}]+\boldsymbol{\mu}],$$
>
> where the terms are computed based on a learned surrogate for the chance constraint and the posterior mean $\boldsymbol{\mu}_{0|t}$.
>
> In contrast, [Guo2024] introduces a Look-Ahead Guidance term designed for linear objectives:
>
> $$\boldsymbol{G}_{t}^{(3)} = -\beta(t)\nabla(y-\boldsymbol{g}^\top \hat{\mathbb{E}}[\boldsymbol{x}_0|\boldsymbol{x}_t])^2,$$
>
> where $\beta(t)$ and $y$ are tuning parameters, $\boldsymbol{g}$ is the gradient of the linear objective, and $\hat{\mathbb{E}}[\boldsymbol{x}_0|\boldsymbol{x}_t] = \alpha^{-1}(t)(\boldsymbol{x}_t + h(t)\boldsymbol{s}(\boldsymbol{x}_t,t))$. This approach is effective when the data distribution is Gaussian and the objective is linear, but may degrade under nonlinear or non-Gaussian scenarios.
>
> Our experimental results further demonstrate that for nonlinear objectives, GGDOpt consistently outperforms the Look-Ahead Guidance from [Guo2024] in terms of both objective value (fval) and computational efficiency (sampling time).
>
> **Table:Comparison results with Look-Ahead Guidance [Guo2024].**
> |Method($\rho=0.1$)| |$n=2$||$n=4$||$n=8$||$n=16$||
> |---|---|---|---|---|---|---|---|---|---|
> | | |fval|time|fval|time|fval|time|fval|time|
> |SOC_CVX| |-0.4558|0.2148|-0.5630|0.2415|-0.6586|0.3214|-0.7394|0.4067|
> |GGDOpt(First-Order)| |-0.4552|0.0156|-0.5615|0.0238|-0.6483|0.0486|-0.7281|0.0614|
> |GGDOpt(Second-Order)| |-0.4558|0.0157|-0.5624|0.0242|-0.6491|0.0507|-0.7372|0.0653|
> |Look-Ahead Guidance[Guo2024]| |-0.4460|0.0329|-0.5181|0.0738|-0.5783|0.1127|-0.6584|0.1436|
>
> As shown above, GGDOpt consistently achieves lower objective values and is approximately $2\times$ faster than the Look-Ahead Guidance across all $n$. This performance gain stems from the computational overhead of [Guo2024], where computing $\boldsymbol{G}_{t}^{(3)}$ requires backpropagation through the score network. In contrast, our guidance terms are derived analytically and thus do **not require any additional gradient computations through the network**, making our method more efficient and scalable.
>
> **Reply to Weakness 4:**
>
> Thank you for raising this valuable point regarding the computational complexity of our method. We agree that diffusion models are generally associated with high training costs. However, in the proposed GGDOpt framework, **the overall complexity is dominated by the training stage**, while the sampling stage is relatively lightweight, especially in the context of large-scale problems.
>
> To support this, we report the computational time for each stage. As shown below, the training time grows significantly with increasing dimension $n$. However, both data generation and sampling time remain negligible by comparison.
>
> **Table:Computational time of three stages(in hours).**
> |Metric||Linear chance constrained problem|||
> |---|---|---|---|---|
> |||$n=8$|$n=16$|$n=128$|
> |Total generating time||0.03|0.06|0.11|
> |Training time||0.53|0.96|11.64|
> |Total sampling time|First-Order|0.0013|0.0017|0.0057|
> ||Second-Order|0.0014|0.0018|0.0063|
>
> This observation confirms that, although training a diffusion model is costly, once trained, our method enables efficient sampling, making it suitable for large-scale CCPs.
>
> Once again, we thank you for your thorough evaluation and thoughtful suggestions. Your feedback has been immensely helpful in strengthening the presentation and scope of the paper.
>
> The corresponding references are listed in order.
>
> 1.[Guo2024] Guo, Yingqing, et al. Gradient guidance for diffusion models: An optimization perspective. Advances in Neural Information Processing Systems, 2024.
>
> 2.[Ma2019]  Ma, Yi-An, et al. Sampling can be faster than optimization. Proceedings of the National Academy of Sciences 116.42, 2019.

---

> ### Author Response · Authors · 2025-08-06
>
> **reply**
>
> Many thanks for your detailed and insightful comments. We appreciate the opportunity to clarify the theoretical justification of our convergence result in Theorem 2 and the implications of score approximation errors.
>
> **Reply to Comment 1: Convergence in Theorem 2.**
>
> We politely disagree with the claim that Theorem 2 is **incorrect**. The concern seems to stem from a misunderstanding between Theorem 1 and Theorem 2 in [Pidstrigach2022].
>
> Our proof is not based on Theorem 1 of [Pidstrigach2022], which assumes access to the true score function. Instead, our proof **invokes Theorem 2** in [Pidstrigach2022], which explicitly analyzes the reverse-time SDE with **an estimated score function**. This result establishes that, under a bounded drift error, the Girsanov weight sequence $(Z_t)$ is a uniformly integrable martingale. Consequently, the marginal distribution of the reverse process using the approximated score is equivalent to that obtained using the true score.
>
> Our Lemma 1 resembles Theorem 1 in structure, but it is derived using an estimated score function from the beginning. We are sorry that this may have caused confusion and we will revise the appendix to more clearly explain this distinction and explicitly reference Theorem 2 from [Pidstrigach2022] as the basis of our proof.
>
> **Reply to Comment 2: On Equation (21) and the estimation of the score.**
>
> Yes, the true distribution of $x_0$ is generally unavailable in practice. As you mentioned, in our implementation, we approximate the expectation in (16) by using empirical samples from the training dataset and solve the optimization via standard methods (e.g., Adam or SGD). However, Equation (21) requires a bounded error in the data distribution $p_0$, not the true distribution of $x_0$. This follows from the definition of $\tilde{p} _ 0$ as the product of the data distribution $p_0$ and the Boltzmann distribution $B_{\beta}$. After training, the error between the score estimator and the true score is much smaller, then the error is bounded, as formalized in Equation (21).
>
> Our claim does not say that Equation (21) holds for arbitrary $\delta$, but rather for **existing** a $\delta$,	through sufficient training. We will update the discussion to clarify that our result is conditional on Assumption (21).
>
> **Reply to Comment 3: Guarantee of chance constraint by inaccurate estimation of the distribution.**
>
> We agree that the exact satisfaction of chance constraints depends on the quality of the score function. This is why we have included both theoretical and empirical assessments of feasibility in the revised manuscript:
>
> -**Theoretically,** we show that under a bounded score error, the sampling distribution converges to one supported within the feasible region.
>
> -**Empirically,** as shown in Section 5.2 and (Table: Empirical feasibility of different algorithms under some uncertainty levels in the rebuttal), we evaluate the estimated constraint satisfaction probability using Monte Carlo simulations and show that the generated solutions maintain feasibility across multiple problem settings.
>
> Thank you for your careful reading and helpful suggestion regarding the notation. We agree that the symbol $\boldsymbol{\theta}$ in Equations (74) and (80) refers to different quantities and may cause confusion. In the revised version, we will adopt different notations to distinguish them clearly and avoid ambiguity.
>
> The corresponding reference is listed below.
>
> 1. [Pidstrigach2022] Pidstrigach, Jakiw. Score-based generative models detect manifolds. Advances in Neural Information Processing Systems, 2022.

---

> > ### Comment · Reviewer_u5CA · 2025-08-06
> >
> > Thank you very much for guiding me to read Theorem 2 in [Pidstrigach2022].  Please note that in [Pidstrigach2022], "If two measures have the same support, they are said to be equivalent" (a statement in Page 3). Therefore, if the score matching error is bounded, the sampled distribution and the original distribution have the same support but they can be different. Did you miss this point?

---

> > > ### Author Response · Authors · 2025-08-07
> > >
> > > **reply**
> > >
> > > Thank you for your insightful comment. We agree that Theorem 2 in [Pidstrigach2022] only ensures that the sampling distribution shares the same support as the true distribution when the score error is bounded, and they may still differ.
> > >
> > > **We did not miss this point.** On the contrary, our proof of Theorem 2 carefully incorporates this distinction. For example, we explicitly highlight in the proof of Theorem 2 in Line 343 that **the sampling distribution of GGDOpt will attain the  same support as the data distribution**.
> > >
> > > To further clarify this in our response and revised manuscript, we offer a brief sketch of the argument used in our proof of Theorem 2:
> > >
> > > --**Support equivalence.**
> > >
> > > Based on Theorem 2 in [Pidstrigach2022], if the score matching error is bounded and the number of diffusion steps satisfies $T \to \infty$, then the sampling distribution $p _ {\text{sample}}(\boldsymbol{x} _ 0|\rho)$ will have the same support as the data distribution $\tilde{p} _ {0}(\boldsymbol{x} _ 0|\rho)\propto p _ {0}(\boldsymbol{x} _ 0|\rho)B _ {\beta}(\boldsymbol{x} _ 0)$, where $B _ {\beta} (\boldsymbol{x} _ 0) \propto e^{-\beta f(\boldsymbol{x} _ 0)}$ is the Boltzmann distribution.
> > >
> > > --**Support of $\tilde{p}_0$ remains $\mathcal{D}\rho$.**
> > >
> > > Since $p_0(\boldsymbol{x}_0|\rho)$ has support $\mathcal{D}\rho$ and the Boltzmann factor only changes the relative density within that domain, the support of $\tilde{p}_0$ also remains $\mathcal{D}\rho$. Thus,
> > >
> > > $$\text{supp } p _ {sample}(\boldsymbol{x} _ 0|\rho) = \text{supp } \tilde{p} _ {0}(\boldsymbol{x} _ 0|\rho)=D _ {\rho}.$$
> > >
> > > --**Letting $\beta \to \infty$ gives optimizer.**
> > >
> > > As $\beta \to \infty$, sampling from $\tilde{p} _ 0$ is equivalent to solving the optimization problem $\boldsymbol{x} ^ * = \text{argmin} _ {\boldsymbol{x} \in \mathcal{D} _ {\rho}} f(\boldsymbol{x})$. Therefore, as both $T \to \infty$ and $\beta \to \infty$, the sample $\tilde{\boldsymbol{x}} _ T$ concentrates around $\boldsymbol{x}^*$.
> > >
> > > We also acknowledge your observation that Equation (83) in our appendix may be misleading by suggesting that two distributions are equal. We will revise it to reflect that what is guaranteed is support equivalence, as in the corrected version above.
> > >
> > > The corresponding reference is listed below.
> > >
> > > 1. [Pidstrigach2022] Pidstrigach, Jakiw. Score-based generative models detect manifolds. Advances in Neural Information Processing Systems, 2022.

---

> > > > ### Comment · Reviewer_u5CA · 2025-08-07
> > > >
> > > > Thank you for the clarification. The statement under the proof of Theorem 2 makes sense but as you mentioned, Eqn. (83) is not rigorous. I will increase the score conditioning on a thorough and rigorous revise of Theorem 2 and its proof and an explanation on how the condition (21) in Theorem 2 is satisfied.

---

> > > > > ### Author Response · Authors · 2025-08-08
> > > > >
> > > > > **reply**
> > > > >
> > > > > We sincerely thank you for the valuable comments and constructive feedback, which have greatly helped us improve our work. Based on the above discussion, we will revise the statement of our Lemma 1 and Theorem 2, and supplement more insightful and thorough explanation in the context.
> > > > >
> > > > > **Assumption 4.** For the forward process
> > > > >
> > > > > $$d\boldsymbol{x}_t= \boldsymbol{a}(\boldsymbol{x}_t,t)dt+b(t) d\boldsymbol{B}_t,$$
> > > > >
> > > > > there is a constant $C$ such that
> > > > >
> > > > > (i) $\boldsymbol{a}(\boldsymbol{x}_t,t)$ is globally Lipschitz for any $t\in[0,T]$, i.e. $\|\boldsymbol{a}(\boldsymbol{x}_t,t)-\boldsymbol{a}(\boldsymbol{x}_t^{\prime},t)\|\leq C\|\boldsymbol{x}-\boldsymbol{x}_t^{\prime}\|$;
> > > > >
> > > > > (ii) $\boldsymbol{a}(\boldsymbol{x}_t,t)$ grows at most linearly for any $t\in[0,T]$, i.e. $\|\boldsymbol{a}(\boldsymbol{x}_t,t)\|\leq C(1+\|\boldsymbol{x}_t\|)$;
> > > > >
> > > > > (iii) $\boldsymbol{x}_t$ has a density $p_t\in\mathcal{C}^1$ for every $t>0$ and
> > > > >
> > > > > $$\int_{t_{0}}^{1}\int_{\|\boldsymbol{x}_t\|<R} |p _ {t}(\boldsymbol{x}_t)|^{2}+|\nabla p _ {t}(\boldsymbol{x}_t)|^{2}dxdt<\infty,$$
> > > > >
> > > > > for any $R>0$ and $0 <t_0\leq T$;
> > > > >
> > > > > (iv) For each $S\in(0, T)$ and all $\|\boldsymbol{x}_t\|\leq N_R$ and $\|\boldsymbol{x}_t^{\prime}\|\leq N_R$, there is a constant $C _ {S,N_R}$ such that $\nabla\log p_t(\boldsymbol{x}_t)$ is locally Lipschitz, i.e.,
> > > > >
> > > > > $$\|\nabla \log p_t(\boldsymbol{x} _ t)-\nabla\log p_t(\boldsymbol{x} _ t^{\prime})|\leq C _ {S,N_R}|\boldsymbol{x}_t-\boldsymbol{x}_t^{\prime}\|,$$
> > > > >
> > > > > for all $t\in(S, T)$.
> > > > >
> > > > > **Remarks on Assumption 4.** Conditions (i)-(iii) are technical conditions on the forward SDE. They ensure that if we run a solution $p_t (\boldsymbol{x} _ t)$ to the forward SDE, then $p _ {T-t}(\boldsymbol{x} _ {T-t})$ will be a solution to the reverse SDE. The last condition ensures that the solutions to the reverse SDE are unique. Assumption 4 can be expected to hold in practice, i.e., for any affine $\boldsymbol{a}(\cdot,t)$ and bounded data manifold.
> > > > >
> > > > >
> > > > > **Lemma 1 (Theorem 2 of [pidstrigach2022]).**
> > > > >
> > > > > Given a forward SDE with marginals $p_t(\boldsymbol{x} _ t)$ and an approximated score $\boldsymbol{s} _ {\boldsymbol{\theta}}(\boldsymbol{x} _ t,t)$ to $\nabla \log p_t(\boldsymbol{x} _ t)$, if the approximation error $\|\boldsymbol{s} _ {\boldsymbol{\theta}}(\boldsymbol{x}_t,t)-\nabla \log p_t(\boldsymbol{x}_t)\|$ is bounded and Assumption 4 holds, then the marginal distribution of the reverse process using the approximated score starting from $p_T(\boldsymbol{x}_T)$ will have the same support as the data distribution $p_0(\boldsymbol{x}_0)$.
> > > > >
> > > > > **Theorem 2.** For any given $\rho\in(0,1)$, suppose that there exists a constant $\delta$ such that the error in the score estimation can be bounded as:
> > > > >
> > > > > $$\|\tilde{\boldsymbol{s}} _ {\boldsymbol{\theta}}(\boldsymbol{x} _ t,t,\rho) + \boldsymbol{G} _ t - \nabla \log \tilde{p} _ t(\boldsymbol{x} _ t|\rho)\| \leq \delta, \quad \forall~\boldsymbol{x}_t.$$
> > > > >
> > > > > For samples $\tilde{\boldsymbol{x}} _ {sample}\sim p _ {sample}(\boldsymbol{x}_0|\rho)$ generated by the reverse process
> > > > >
> > > > > $$d\boldsymbol{x} _ t=[\boldsymbol{a}(\boldsymbol{x} _t,t)-b(t)^2(\tilde{\boldsymbol{s}} _ {\boldsymbol{\theta}}(\boldsymbol{x}_t,t,\rho)+\boldsymbol{G}_t )]dt+b(t)d\boldsymbol{\bar{B}} _t,$$
> > > > >
> > > > > with prior $p_{prior} = \mathcal{N}(\boldsymbol{0},\boldsymbol{I})$, affine drift coefficients $\boldsymbol{a}(\cdot,t)$, and
> > > > >
> > > > > $$\tilde{\boldsymbol{s}} _ {\boldsymbol{\theta}}(\boldsymbol{x} _ t,t,\rho)= (1+w) \boldsymbol{s} _ {\boldsymbol{\theta}}(\boldsymbol{x}_t,t,\rho) -w\boldsymbol{s} _ {\boldsymbol{\theta}}(\boldsymbol{x}_t,t,\emptyset),$$
> > > > >
> > > > > as $T\rightarrow \infty$, $p _ {sample}(\boldsymbol{x} _0|\rho)$ will have the same support as $\tilde{p} _ {0}(\boldsymbol{x} _ 0|\rho)$. Further, as $\beta\rightarrow\infty$, $\tilde{\boldsymbol{x}} _ {sample}$ will concentrate around $\boldsymbol{x} ^ * = \text{argmin} _ {\boldsymbol{x}\in\mathcal{D} _ {\rho}} f(\boldsymbol{x})$.

---

> > > > > ### Author Response · Authors · 2025-08-08
> > > > >
> > > > > **Proof of Theorem 2.**
> > > > >
> > > > > For the forward process $d\boldsymbol{x} _ t = \boldsymbol{a}(\boldsymbol{x} _ t,t)dt+b(t) d\boldsymbol{B} _ t, t\in [0,T]$ with affine drift coefficients $\boldsymbol{a}(\cdot,t)$, conditions (i)-(ii) in Assumption 4 are satisfied. For the given data set $\{\boldsymbol{x} ^ {(i)}\} _ {i=1}^{N}$ contained in a ball of radius $M_R$, we have that $\log \tilde{p} _ t(\boldsymbol{x} _ t,t)\in \mathcal{C} ^ {\infty}$ in both $t$ and $\boldsymbol{x} _ t$ for $t>0$ where the product distribution $\tilde{p} _0(\boldsymbol{x} _ 0|\rho) \propto p _ {0}(\boldsymbol{x} _ 0|\rho)B _ {\beta}(\boldsymbol{x}_0)$. Therefore we can integrate $\tilde{p} _ t$ and its derivative over compact sets, implying that condition (iii) holds. Furthermore, for each $S\in(0, T)$, the Hessian w.r.t. $(\boldsymbol{x} _ t, t)$ is continuous and obtains its maximum and minimum on the compact set $[S,T]\times B _ {N_R}$, where $B _ {N_R}$ is the ball of diameter $N_R$ around the origin. Therefore, the gradient $\nabla\log \tilde{p} _ t(\boldsymbol{x}_t)$ is Lipschitz on $[S,T]\times B _ {N_R}$, which proves condition (iv).
> > > > >
> > > > > The stationary distribution of the forward process is characterized by the corresponding Fokker-Planck equations, where $p_T = \mathcal{N}(\boldsymbol{0},\boldsymbol{I})$ when $T\rightarrow\infty$. Then we have that $p_{prior} = p_T$. Based on Lemma 1, if the score matching error is bounded, then the sampling distribution $p _ {sample}(\boldsymbol{x} _ 0|\rho)$ with prior $p _ {prior} = \mathcal{N}(\boldsymbol{0},\boldsymbol{I})$ will have the same support as the product distribution $\tilde{p} _ 0(\boldsymbol{x} _ 0|\rho)\propto p _ {0}(\boldsymbol{x} _ 0|\rho)B _ {\beta}(\boldsymbol{x} _ 0)$, where $B _ {\beta}$ is the Boltzmann distribution $B _ {\beta}(\boldsymbol{x} _0)\propto e^{-\beta f(\boldsymbol{x} _ 0)}$.
> > > > >
> > > > > Since $p _ {0}(\boldsymbol{x} _ 0|\rho)$ has support $\mathcal{D} _ {\rho}$ and the Boltzmann factor only changes the relative density within that domain, the support of $\tilde{p} _ 0(\boldsymbol{x}_0|\rho)$ also remains $\mathcal{D} _ {\rho}$, i.e.,
> > > > >
> > > > > $$\text{supp } p _ {sample}(\boldsymbol{x} _ 0|\rho)=\text{supp } \tilde{p} _ {0}(\boldsymbol{x} _ 0|\rho) = \mathcal{D} _ {\rho}.$$
> > > > >
> > > > > As $\beta \rightarrow \infty$, sampling from the product distribution $\tilde{p} _ 0(\boldsymbol{x} _ 0|\rho)$ is equivalent to solving the optimization problem $\boldsymbol{x}^* = \text{argmin} _ {\boldsymbol{x}\in\mathcal{D} _ {\rho}} f(\boldsymbol{x})$. Then we have that as $T \rightarrow \infty$ and $\beta \rightarrow \infty$, the sample $\tilde{\boldsymbol{x}} _ {sample}$ will concentrate around $\boldsymbol{x}^*$.
> > > > >
> > > > > Theorem 2 establishes that, by introducing an additional gradient guidance term into the reverse process, the sampling distribution of GGDOpt will attain the exact same support as the data distribution. Moreover, as the inverse temperature parameter $\beta$ increases, the sampling distribution becomes increasingly concentrated around points with the lowest function values within the support of the data distribution.
> > > > >
> > > > > **Explanation on Equation (21).**
> > > > >
> > > > > The assumption in Equation (21) quantifies the approximation accuracy of the trained score network relative to the true score function. We acknowledge that this assumption can be difficult to verify directly in practice, as it depends on the training quality of the neural network and the expressiveness of the model class. However, this type of assumption is commonly used in the theoretical analysis of diffusion models (see, e.g., [Pidstrigach2022], [De2021], [De2022]) and is used to establish convergence results in generative modeling and sampling.
> > > > >
> > > > > During the training phase, we approximate the expectation in (16) by using empirical samples from the training dataset and solve the optimization via standard methods (e.g., Adam or SGD). After training, the error between the score estimator and the true score is much smaller, then the error may be bounded, which implies that (21) holds.
> > > > >
> > > > > The corresponding reference is listed below.
> > > > >
> > > > > 1. [Pidstrigach2022] Pidstrigach, Jakiw. Score-based generative models detect manifolds. Advances in Neural Information Processing Systems, 2022.
> > > > >
> > > > > 2. [De2021] De Bortoli, Valentin, et al. Diffusion schrödinger bridge with applications to score-based generative modeling. Advances in Neural Information Processing Systems, 2021.
> > > > >
> > > > > 3. [De2022] De Bortoli, Valentin. Convergence of denoising diffusion models under the manifold hypothesis. arXiv preprint arXiv:2208.05314, 2022.

---

### Official Review · Reviewer_fiiv · 2025-07-02

**Clarity:** 4
**Significance:** 3
**Originality:** 3
**Rating:** 4
**Confidence:** 3

**Summary:**

This paper addresses a broad and important class of optimization problems—those governed by probabilistic constraints. By reformulating these problems as sampling tasks (nonconvex CCP with intractable constraints <-> sampling from an unknown distribution), the authors enable the use of powerful modern generative models from machine learning, such as diffusion models, which can learn to sample effectively from the feasible or high-quality regions of complex optimization landscapes. The approach is supported by both theoretical guarantees and empirical results.

**Questions:**

First, it would be helpful for the paper to engage more directly with the existing literature on diffusion-guided optimization, particularly in relation to the setup described in lines 62–67. This setting appears to be fairly common in recent work (e.g.,  https://arxiv.org/abs/2404.14743 or https://proceedings.mlr.press/v202/krishnamoorthy23a.html). A discussion of this growing body of literature would provide useful context and help clarify the novelty and significance of the current contribution. Additionally, in lines 141–144, the paper states that it uses Tweedie’s formula for the mean and introduces a free parameter for the variance. This seems like a substantial simplification. Could the authors clarify this choice, especially since the variance can also be derived from Tweedie’s formula? If the Hessian is already being computed for the guidance term, it’s unclear why the variance is treated as an independent parameter. Apologies if I missed this discussion elsewhere—the implementation details in Section 3.3 don’t appear to address it.

Secondly, Equation (15) includes a final parameter whose role isn’t clearly explained in the manuscript—I couldn’t find it defined or referenced elsewhere. It would be helpful to clarify what this parameter represents and how it is chosen or tuned. Additionally, Assumption 1 is presented without much accompanying intuition. A brief discussion of its implications would make the theoretical framework more accessible, especially for readers unfamiliar with the nuances of these types of probabilistic constraints. Providing some intuition or examples to illustrate what the assumption means in practice would greatly improve clarity.

**Ethical Concerns:**

["NO or VERY MINOR ethics concerns only"]

**Final Justification:**

Thanks for the rebuttal, especially around the discussion of Hessian. I (and the other reviewers) are still concerned about the lack of adequate related work and also about paper clarity. Therefore I maintain my score.

**Limitations:**

Yes

**Quality:**

3

**Strengths And Weaknesses:**

The paper presents a compelling approach to an important class of optimization problems with wide-ranging applications. The writing is generally clear and accessible, making the paper enjoyable to read. The method is well-motivated, and the integration of modern generative models like diffusion into constrained optimization is both novel and promising. However, Figure 1—introduced early in the paper—misses an opportunity to clearly explain the method. Its complexity makes it difficult to interpret, and it would benefit from simplification and a more detailed caption that guides the reader on how to read and understand it.

On the other hand, several concerns relate to computational cost and experimental evaluation. The paper lacks sufficient discussion of the computational overhead associated with the proposed approach—for instance, the practicality of using the Hessian (as in Equation 12) within a diffusion model framework remains unclear. Additionally, the cost of generating training data in Stage 1 may be significant, and it's important to understand how this compares to the total runtime of the optimization pipeline. The experimental section also raises questions: the use of small problem sizes (e.g., n=8n=8) limits the practical relevance of the results, and the reported runtimes are too short to showcase meaningful performance differences. It’s also unclear whether Table 1 reflects performance across the full optimization run or only at the final iteration. The results in Table 2 are similarly underwhelming; the paper would be strengthened by including more demanding experimental setups with larger problem instances and longer runtimes to better demonstrate the method’s capabilities.

---

> ### Author Rebuttal · Authors · 2025-07-31
>
> **Reply**
>
> We sincerely thank you for your thoughtful and constructive feedback, which allows us to clarify and further improve our work. Below we provide a point-by-point response to each of your comments.
>
> **Reply to Weakness 1:**
>
> We appreciate your suggestion on Figure 1. To help readers better understand the framework, we provide an additional explanation in the caption of Figure 1 as follows:
>
> >**Figure 1:**Overview of the GGDOpt framework.
> >(1)Generate a training set of points satisfying the chance constraint by solving a deterministic restricted problem.  (2)Train a diffusion model with classifier-free guidance to learn the score of the conditional distribution.  (3)Perform the reverse diffusion process with additional gradient guidance to sample from the product of the data distribution and the Boltzmann distribution.
>
> **Reply to Weakness 2:**
>
> Thank you for your valuable comments regarding the computational cost and evaluation. We test the linear chance constrained problem with $n=8$ and repeat 100 times to calculate the empirical mean of the objective value (fmean), the empirical standard deviation (fstd), and the average sample time (time). The results are summarized in the following table.
>
> **Table:Comparison of the proposed methods.**
> |Method|SOC_CVX|GGDOpt(First-Order)|GGDOpt(Second-Order)||||
> |---|---|---|---|---|---|---|
> ||||$\beta=0.1$|$\beta=1$|$\beta=10$||
> |fmean|-0.6586|-0.6483|-0.6341|-0.6548|-0.6585|
> |fstd|0|0.0051|5.6726e-3|2.5112e-05|2.2329e-08|
> |time|0.3214|0.0486|0.0569|0.0527|0.0541|
>
> As observed, the second-order method achieves lower objective values compared to the first-order method and its performance closely matches the optimal solution obtained by SOC\_CVX. Moreover, the second-order method leads to significantly lower standard deviations, particularly as $\beta$ increases.
>
> Regarding the computational overhead, we acknowledge that in general high-dimensional settings, inverting a full Hessian matrix can be costly. However, in our current experiments, the objective is quadratic and its Hessian is diagonal which makes the inversion trivial.
>
> To summarize, when the problem dimension is modest and the Hessian matrix has a simple structure (e.g., diagonal or sparse), the second-order guidance is both practical and beneficial. In contrast, for high-dimensional problems with dense and expensive-to-invert Hessians, the first-order guidance remains a computationally efficient alternative. We will clarify this trade-off and the underlying assumptions in the revised manuscript.
>
> **Reply to Weakness 3:**
>
> You are absolutely right that the cost of generating training data in Stage 1 is crucial. In our framework, training samples are generated by solving the relaxed problem (RP), which is typically much easier to solve than the original CCP. Here we provide the costs of three stages for the linear chance constraint problem in Section 5.1. For each $n$, we generate 1000 data in the training stage. During sampling, we execute 100 times of reverse process to analyze the stability of GGDOpt. The total time costed in hour is shown below.
>
> **Table:Computational time of three stages(in hours).**
> |Stage||Linear chance constrained problem|||
> |---|---|---|---|---|
> |||$n=8$|$n=16$|$n=128$|
> |Data generating time||0.03|0.06|0.11|
> |Training time||0.53|0.96|11.64|
> |Total sampling time|First-Order|0.0013|0.0017|0.0057|
> ||Second-Order|0.0014|0.0018|0.0063|
>
> Furthermore, our experiments indicate that increasing the quantity of training data alone does not guarantee better performance. Instead, high-quality samples closer to the true optimal solutions are the key of effective guided sampling.
>
> **Reply to Weakness 4:**
>
> Thank you for highlighting the limitations regarding problem scale and runtime. We acknowledge that Tables 1 and 2 only report the runtime of **the sampling stage**, excluding the computational cost of data generation and model training.
>
> To better demonstrate the scalability and applicability of GGDOpt, we have conducted additional experiments on much larger problem instances, increasing the problem dimension $n$ up to 1024. The results are summarized in the table below.
>
> **Table:Experimental results with different $n$.**
> |Method($\rho=0.1$)|$n=8$||$n=16$||$n=128$||$n=1024$||
> |---|---|---|---|---|---|---|---|---|
> | |fval|time|fval|time|fval|time|fval|time|
> |SOC_CVX|-0.6586|0.3214|-0.7394|0.4067|-0.8951|0.8751|-0.9610|1.2483|
> |SAA_PDCA|-0.6301|0.4572|-0.6686|0.6530|-0.8582|1.7348|-0.8956|6.4432|
> |GGDOpt_FirstOrder|-0.6483|0.0486|-0.7281|0.0614|-0.8836|0.2068|-0.9409|0.4258|
> |GGDOpt_SecondOrder|-0.6491|0.0507|-0.7372|0.0653|-0.8941|0.2269|-0.9547|0.4699|
>
> As expected, the runtime increases with the problem dimension. However, both the first- and second-order versions of GGDOpt remain consistently faster than the baseline SAA\_PDCA method across all dimensions. Moreover, the increase in runtime is moderate, indicating that our approach scales favorably even in high-dimensional settings.
>
> **Reply to Question 1:**
>
> Thanks for recommending these two related papers. In [Krishnamoorthy2023], a conditional diffusion model is trained using loss reweighting to map function values to high-quality samples, and applied to offline black-box optimization. In [Guo2024], a Look-Ahead Guidance mechanism is introduced to preserve the linear structure of the data, enabling applications to regularized and global optimization in settings where the data distribution is Gaussian and the objective is linear.
>
> In contrast, our work is, to the best of our knowledge, the first to apply diffusion models to general chance constrained problems. The key challenge is the lack of readily available training data corresponding to the product distribution formed by the objective function and the chance constraint. We address this by introducing a dedicated data generation stage, where high-quality feasible samples are obtained, followed by conditional training of a score-based diffusion model.
>
> Furthermore, we propose a gradient-guided reverse process to sample from the product distribution. Our guidance terms are derived in closed form based on the structure of the product distribution, which has a different form in contrast to the guidance term in [Guo2024]. In addition, the proposed guidance term does not require backpropagation through the score network.
>
> **Reply to Question 2:**
>
> We apologize for the earlier ambiguity. While Tweedie’s formula theoretically provides both the posterior mean and covariance, $\boldsymbol{\Sigma}_{0|t}=(1-\bar{\alpha}_t)(\boldsymbol{I}+(1-\bar{\alpha}_t)\nabla^2\log p(\boldsymbol{x}_t))$, computing the covariance requires evaluating the Hessian of $\log p(\boldsymbol{x})$.
>
> In our framework, the score function $\boldsymbol{s}_{\boldsymbol{\theta}}$ is parameterized by a neural network, and computing its second derivatives involves backpropagation through the network’s Jacobian, which is computationally expensive, especially in high dimensions.
>
> To strike a balance between performance and efficiency, we choose to treat the covariance as a tunable constant. We acknowledge that this introduces an approximation, but as shown below, this achieves comparable objective values to the fully Tweedie-based method, while reducing runtime by more than an order of magnitude. These results confirm that using a fixed variance can be a practical and robust alternative.
>
> **Table:Experimental results with different variance schedules.**
> |$n=8,\rho=0.1$|Tweedie's $\boldsymbol{\Sigma}$|GGDOpt(second-order)|||||
> |---|---|---|---|---|---|---|
> | | |$\sigma=0.01$|$\sigma=0.02$|$\sigma=0.1$|$\sigma=1$|$\sigma=10$|
> |fval|-0.6571|-0.6471|-0.6457|-0.6545|-0.6320|-0.6049|
> |time|1.0984|0.0491|0.0496|0.0493|0.0492|0.0493|
>
> **Reply to Question 3:**
>
> Thank you for highlighting the unclear notation. The final parameter $\emptyset$ denotes the absence of conditioning information, following the classifier-free guidance scheme. Specifically, during training, we randomly omit the conditioning variable $\rho$ so that the model learns both conditional and unconditional score estimates. At inference time, this allows us to construct guided samples via a linear interpolation of these two estimates, as originally proposed in [Ho2022].
>
> **Reply to Question 4:**
>
> We appreciate your feedback regarding Assumption 1. The first and second assumptions in Assumption 1 describe the forward and reverse processes of an Ornstein–Uhlenbeck (OU) process, respectively; such formulations are standard in the diffusion literature for analyzing the impact of the finite diffusion time on the convergence (e.g., [Song2020]).
>
> The third assumption imposes **a growth bound on the gradient of the objective function**. This type of regularity condition is common in the convergence analysis of stochastic optimization and sampling algorithms, particularly when studying stability and convergence under Langevin dynamics or diffusion-based methods (see, e.g., [Raginsky2017]). In practice, this assumption holds for a broad class of functions, including smooth bounded functions and quadratic objectives, which frequently arise in real-world optimization problems.
>
> The corresponding references are listed in order.
>
> 1.[Krishnamoorthy2023] Krishnamoorthy, Siddarth, et al. Diffusion models for black-box optimization. International Conference on Machine Learning. PMLR, 2023.
>
> 2.[Guo2024] Guo, Yingqing, et al. Gradient guidance for diffusion models: An optimization perspective. Advances in Neural Information Processing Systems, 2024.
>
> 3.[Ho2022] Ho, Jonathan, and Tim Salimans. Classifier-free diffusion guidance. arXiv preprint arXiv:2207.12598, 2022.
>
> 4.[Song2020] Song, Yang, et al. Score-based generative modeling through stochastic differential equations. arXiv preprint arXiv:2011.13456, 2020.
>
> 5.[Raginsky2017] Raginsky, Maxim, et al. Non-convex learning via stochastic gradient langevin dynamics: a nonasymptotic analysis. Conference on Learning Theory. PMLR, 2017.

---

> > ### Comment · Reviewer_fiiv · 2025-08-07
> >
> > Thanks for this rebuttal. I remain positive about this work, although somewhat concerned by the theoretical issues raised by the other reviewer. The paper is undoubtedly stronger now that you have ran all this additional ablation (timings + robusts of params + extra comparisons e.g. with MIP). All the reviewers have suggested that you are missing key related works, and this needs to be addressed in the final version.
> >
> > Reply to Weakness 1: Thanks this is clearer
> > Reply to Weakness 2: I now understand that the Hessian is accessible for your particular problems (and those with other particular structures). Please make sure to elaborate on this in the final version, as I missed this the first time I read it (this is probably my fault…).
> > Reply to Weakness 3: Thanks. Do you have any intuition about how similar to the true problem a “relaxed problem” would have to be to ensure that the diffusion model can be used to solve it using your framework? I guess this poses an interesting trade-off between cost of data acquisition (i.e. level of approx solution for training data) v.s. Quality of end solution after your method?
> > Reply to Weakness 4: This is very reassuring thanks!
> > Reply to Question 2:  Thanks, this is what I assumed you were doing. Please make this clearer in the final version.
> > Reply to Question 4: Thanks for this. It makes it easy for those not experts in this subfield.

---

> ### Author Response · Authors · 2025-08-08
>
> **reply**
>
> We sincerely thank you for your valuable comments and constructive feedback, which have significantly strengthened our work. Following the reviewers' suggestions, we will:
>
> 1. include a more comprehensive discussion of key related works in the final version,
>
> 2. provide a detailed explanation of our framework in Figure 1 and the Hessian's role, and
>
> 3. enhance clarity by expanding critical subsections as recommended.
>
> **Data generation via solving RPs.**
>
> You raise an important point regarding the critical role of data generation in our framework. A primary objective of our method is to solve chance constrained problems via solving simple restricted problems. The approach of obtaining high-quality CCP solutions via RP solutions is well-motivated by practical applications. Consider the CCP
>
> $$\min _ {\boldsymbol{x}}\quad f(\boldsymbol{x}) \quad \text{s.t.} \quad\boldsymbol{x}\in\mathcal{X} _ {\rho},$$
>
> where the chance (or probabilistic) constraint set $\mathcal{X} _ {\rho}$ is defined by
>
> $$\mathcal{X} _ {\rho} = (\boldsymbol{x}\in \mathbb{R}^n |\text{Prob} _ {\boldsymbol{h}}(\boldsymbol{g}(\boldsymbol{x},\boldsymbol{h})\geq\boldsymbol{0})\geq1-\rho).$$
>
> This is generally challenging to solve even when $\boldsymbol{g}$ is linear. Notice that in most of cases, it is much easier to solve the deterministic RP
>
> $$\min _ {\boldsymbol{x}}\quad f(\boldsymbol{x})\quad \text{s.t.}\quad \boldsymbol{g}(\boldsymbol{x},\bar{\boldsymbol{h}})\geq\boldsymbol{z}_i,$$
>
> with fixed $\bar{\boldsymbol{h}}$ and $\boldsymbol{z} _ i\geq \boldsymbol{0}$. With the increase of $\boldsymbol{z} _ i$, the confidence that the solution $\boldsymbol{x}(\boldsymbol{z} _ i)$ satisfies the chance constraint increases accordingly. This enables us to generate feasible points within the chance constraint as well as obtaining better objective values for any given risk level $\rho$. Further, under certain regularity conditions, data generated in this way could include the solution to the original CCP. This motivates our data-driven approach to discover the manifold structure of the data points and the CCP solution, which could be learned through diffusion models.
>
> **Trade-off between the cost of data generation and the quality of the returned solution.**
>
> As noted, a trade-off exists between data generation costs and solution quality: higher-quality training data typically yields better approximations but increases computational expense. We observe that when the corresponding RP exhibits favorable structures with efficient solvers, our framework becomes particularly suitable, e.g., for the linear chance constrained problems and robust waveform design. Characterizing the relation between RP solutions to the sample distribution is an interesting topic. In future work, we will conduct a rigorous analysis of this trade-off and theoretically characterize how data quality impacts the final sample distribution.

---

> > ### Comment · Reviewer_fiiv · 2025-08-08
> >
> > Thanks for the discussion. I think it would be nice to include this in the paper!

---

### Official Review · Reviewer_AaBr · 2025-07-02

**Clarity:** 2
**Significance:** 3
**Originality:** 3
**Rating:** 4
**Confidence:** 1

**Summary:**

This paper proposes a novel approach for solving chance-constrained optimization (CCP) problems by integrating diffusion models into the solution process. The core challenge lies in sampling from a product distribution that combines the constraint-related randomness with the Boltzmann distribution of the objective function. To address this, the authors develop a three-stage algorithm: (1) a data generation phase that produces high-quality design points using a deterministic surrogate of the chance constraint, (2) a diffusion-based learning stage that estimates the score of the product distribution, and (3) a guided sampling stage based on the learned score. Numerical experiments on CCP problems demonstrate that the proposed method outperforms existing algorithms in terms of both objective value and feasibility.

**Questions:**

1. Why does the second-order guidance incur no additional cost compared to first-order? Is this due to problem dimension  is small?

2. In what settings is assumption (21) practically valid? It seems somewhat strong.

3. In line 150, the authors write “convexify the nonconvex feasible region.” However, Problem (13) can still be nonconvex. Could the authors clarify what is meant here—perhaps it refers to tightening the constraint conservatively, rather than true convexification?

**Ethical Concerns:**

["NO or VERY MINOR ethics concerns only"]

**Final Justification:**

My concerns have been addressed, and I have raised my score accordingly.

**Quality:**

3

**Strengths And Weaknesses:**

Strengths:

The paper introduces a new framework that incorporates gradient-guided diffusion models into CCP. Integrating gradient-guided diffusion into CCP is, to my knowledge, new and potentially influential for stochastic optimization. The experimental section demonstrates consistent improvements over competitive baselines, suggesting that diffusion-based sampling can be worthwhile to study further.


Weakness:

1. The discussion of gradient guidance in Section 2.2 is somewhat brief. For readers unfamiliar with the topic like me, it would be helpful to provide more explanation or intuition. In particular, it is unclear how much of the guidance mechanism is novel in this work, since similar techniques—such as classifier guidance or energy-based guidance—have been proposed in prior literature (e.g., Guo et al., 2024). A more explicit comparison or positioning relative to these works would clarify the contribution.

2. In section 3.1, the data generation procedure in Section 3.1 is not fully transparent. For instance, the estimation of  $\rho^{(i)}$, using Equation (14) could be explained more clearly. The notation uses a subscript without explicitly defining the indicator function used to approximate the chance constraint.

3. The assumption in Equation (21) appears weak or hard to verify in practice. It would be helpful to explain in what types of applications it tends to hold, or whether the method is robust when it is only approximately satisfied.

References:
Guo, Y., Yuan, H., Yang, Y., Chen, M., & Wang, M. Gradient Guidance for Diffusion Models: An Optimization Perspective. In The Thirty-eighth Annual Conference on Neural Information Processing Systems.


Minor:
Line190: parametered should be parameterized

---

> ### Author Rebuttal · Authors · 2025-07-31
>
> **Reply**
>
> We sincerely thank you for your thoughtful and constructive feedback. We truly appreciate your effort in identifying the key weaknesses and raising important questions, which helped us better clarify the scope, contributions, and implementation details of our work. Below we provide a point-by-point response to each of your comments.
>
> **Reply to Weakness 1:**
>
> We appreciate your insightful feedback on our gradient guidance mechanism in Section 2.2. The core intuition of the gradient guidance in our work is to steer the diffusion sampling trajectory toward regions that yield desirable properties (e.g., lower objective values or improved feasibility), by modifying the score function in a principled way. While guidance techniques have been widely applied in areas such as image generation, black-box optimization, and inverse problems, our contribution does not lie in introducing guidance, but rather in deriving **a novel class of guidance terms based on product distributions**, specifically tailored to chance-constrained optimization problems with unknown distributions.
>
> More specifically, compared to prior work such as [Guo2024], our approach introduces two main innovations:
>
> - **Conditional Training and Applicability Beyond Linear-Gaussian Settings:**  Unlike [Guo2024], which applies guidance to pre-trained unconditional diffusion models and assumes a linear objective with Gaussian data, our framework involves a dedicated data generation process followed by conditional score training. This enables us to address nonlinear and structurally complex chance-constrained problems, where directly sampling from the feasible region is nontrivial.
>
> - **A New Class of Guidance Derived from Product Distributions:** Most existing guided diffusion frameworks follow the general SDE form as follows:
>
> $$d\boldsymbol{x}_t=[\boldsymbol{a}(\boldsymbol{x}_t,t)-b(t)^2(\boldsymbol{s}(\boldsymbol{x}_t,t)+\boldsymbol{G}_t )]dt+b(t)d\boldsymbol{\bar{B}}_t.$$
>
> In our work, we derive two types of guidance terms directly from the product distribution formulation of the target density:
>
> 1. a first-order guidance
>
> $$\boldsymbol{G}_t^{(1)} = - \beta \nabla f(\boldsymbol{x}_t),$$
>
> 2. a second-order guidance
>
> $$\boldsymbol{G}_t^{(2)} = -\frac{1}{\sigma^2}[\boldsymbol{H}^{-1}[(\nabla^2 f(\boldsymbol{x}_t) \boldsymbol{x}_t+\nabla f(\boldsymbol{x}_t)) -\frac{1}{\beta \sigma^2}\boldsymbol{\mu}]+\boldsymbol{\mu}],$$
>
> where the terms are computed based on a learned surrogate for the chance constraint and the posterior mean $\boldsymbol{\mu}_{0|t}$.
>
> In contrast, [Guo2024] introduces a Look-Ahead Guidance term designed for linear objectives:
>
> $$\boldsymbol{G}_{t}^{(3)} = -\beta(t)\nabla(y-\boldsymbol{g}^\top \hat{\mathbb{E}}[\boldsymbol{x}_0|\boldsymbol{x}_t])^2,$$
>
> where $\beta(t)$ and $y$ are tuning parameters, $\boldsymbol{g}$ is the gradient of the linear objective, and $\hat{\mathbb{E}}[\boldsymbol{x}_0|\boldsymbol{x}_t]$ is an approximation of the posterior mean that can be calculated by the score network, i.e., $\hat{\mathbb{E}}[\boldsymbol{x}_0|\boldsymbol{x}_t] = \alpha^{-1}(t)(\boldsymbol{x}_t + h(t)\boldsymbol{s}(\boldsymbol{x}_t,t))$. This approach is effective when the data distribution is Gaussian and the objective is linear, but may degrade under nonlinear or non-Gaussian scenarios.
>
> Our experimental results further demonstrate that for the nonlinear chance constrained programming, the proposed GGDOpt consistently outperforms the Look-Ahead Guidance from [Guo2024] in terms of both objective value (fval) and computational efficiency (sampling time).
>
> **Table:Comparison results with Look-Ahead Guidance [Guo2024].**
> |Method($\rho=0.1$)| |$n=2$||$n=4$||$n=8$||$n=16$||
> |---|---|---|---|---|---|---|---|---|---|
> | | |fval|time|fval|time|fval|time|fval|time|
> |SOC_CVX| |-0.4558|0.2148|-0.5630|0.2415|-0.6586|0.3214|-0.7394|0.4067|
> |GGDOpt(First-Order)| |-0.4552|0.0156|-0.5615|0.0238|-0.6483|0.0486|-0.7281|0.0614|
> |GGDOpt(Second-Order)| |-0.4558|0.0157|-0.5624|0.0242|-0.6491|0.0507|-0.7372|0.0653|
> |Look-Ahead Guidance[Guo2024]| |-0.4460|0.0329|-0.5181|0.0738|-0.5783|0.1127|-0.6584|0.1436|
>
> As shown in the above table, our proposed GGDOpt consistently achieves lower objective values and the performance gap between GGDOpt and Look-Ahead Guidance increases with the problem dimension $n$. In terms of computational efficiency, GGDOpt is approximately $2\times$ faster than the Look-Ahead Guidance across all problem sizes. This performance gain stems from the computational overhead of [Guo2024], where computing the guidance term $\boldsymbol{G}_{t}^{(3)}$ requires backpropagation through the score network to obtain the gradient of the posterior mean $\mathbb{E}[\boldsymbol{x}_0|\boldsymbol{x}_t]$ with respect to $\boldsymbol{x}_t$. In contrast, our first- and second-order guidance terms are derived analytically and thus do **not require any additional gradient computations through the network**, making our method more efficient and scalable.
>
> **Reply to Weakness 2:**
>
> Thank you for this insightful comment regarding the transparency of the data generation procedure and the definition of $\rho^{(i)}$ in Section 3.1. In general, the quantity $\text{Prob}_{\boldsymbol{h}}(\boldsymbol{g}(\boldsymbol{x}(\boldsymbol{z}_i),\boldsymbol{h})\geq\boldsymbol{0})$ is hard to compute since it requires a multidimensional	integration over the distribution of the random variable $\boldsymbol{h}$. Inspired by the Sample Average Approximation (SAA) approach, we estimate this probability using an empirical average based on $L$ i.i.d. realizations of $\boldsymbol{h}$:
>
> $$\text{Prob}_{\boldsymbol{h}}(\boldsymbol{g}(\boldsymbol{x}(\boldsymbol{z}_i),\boldsymbol{h})\geq\boldsymbol{0})\approx \frac{1}{L}\sum\ell^{0|1}(\boldsymbol{g}(\boldsymbol{x}(\boldsymbol{z}_i),\boldsymbol{h}^{(\ell)})):=1-\rho^{(i)} ~~~~(1),$$
>
> where $\ell^{0/1}$ is the element-wise indicator function that returns 1 if all components of the argument vector are positive, and 0 otherwise. To compute $\rho^{(i)}$, we proceed as follows:
>
> 1. For each sampled restriction vector $\boldsymbol{z}_i \geq \boldsymbol{0}$, we solve the corresponding restricted problem, which yields a candidate solution $\boldsymbol{x}(\boldsymbol{z}_i)$.
>
> 2. We then draw $L$ independent realizations $\boldsymbol{h}^{(\ell)}$ from the underlying distribution and evaluate  $\rho^{(i)}$.
>
> According to equation (1), we see that $\boldsymbol{x}(\boldsymbol{z}_i)$ may satisfy the chance constraint with parameter $\rho^{(i)}$. Thus, we take $\boldsymbol{x}(\boldsymbol{z}_i)$ and the corresponding $\rho^{(i)}$ generated in this way as the data set of the next training stage.
>
> **Reply to Weakness 3 and Question 2:**
>
> Thank you for highlighting the concern regarding the practicality of the assumption in Equation (21). We provide further clarification below.
>
> The assumption in Equation (21) quantifies the approximation accuracy of the trained score network relative to the true score function. We acknowledge that this assumption can be difficult to verify directly in practice, as it depends on the training quality of the neural network and the expressiveness of the model class. However, this type of assumption is common in the theoretical analysis of diffusion models (see, e.g., [Pidstrigach2022]; [De2021]) and is used to establish convergence results in generative modeling and sampling.
>
> **Reply to Question 1:**
>
> We fully agree that in high-dimensional settings, computing the inverse of a general Hessian matrix can be computationally expensive.
>
> However, in our current experiments, the objective function is quadratic, which has a diagonal Hessian matrix. As a result, the matrix inversion involved in the second-order guidance becomes trivial and introduces negligible overhead compared to that in the first-order method. This is why, in our reported results, the computational cost of the second-order guidance is almost identical to that of the first-order guidance.
>
> **Reply to Question 3:**
>
> Thank you for your careful reading. You are correct that the restricted optimization problem (13) can indeed be nonconvex. As noted, this subproblem is used in Stage 1 to generate high-quality training samples for the diffusion model, which is not intended as a convex relaxation of the original chance-constrained problem.
>
> Our phrasing "convexify the nonconvex feasible region" refers to the forward diffusion process used in Stage 2, where adding the Gaussian noise to the indicator function of the feasible set gradually transforms it into a log-concave density. The evolving $\epsilon$-support of this smoothed distribution tends to exhibit an increasingly convex geometry, in an approximate sense.
>
> We once again thank you for your insightful feedback. We believe your comments have helped us greatly improve the clarity and rigor of our work, and we hope our revisions and responses address your concerns satisfactorily.
>
> The corresponding references are listed in order.
>
> 1.[Guo2024] Guo, Yingqing, et al. Gradient guidance for diffusion models: An optimization perspective. Advances in Neural Information Processing Systems, 2024.
>
> 2.[Pidstrigach2022] Pidstrigach, Jakiw. Score-based generative models detect manifolds. Advances in Neural Information Processing Systems, 2022.
>
> 3.[De2021] De Bortoli, Valentin, et al. Diffusion schrödinger bridge with applications to score-based generative modeling. Advances in Neural Information Processing Systems, 2021.

---

> > ### Comment · Reviewer_AaBr · 2025-08-05
> >
> > Thank you for the rebuttal. My concerns have been addressed, and I have raised my score accordingly.

---

### Official Review · Reviewer_2s5s · 2025-07-06

**Clarity:** 1
**Significance:** 2
**Originality:** 3
**Rating:** 4
**Confidence:** 2

**Summary:**

Chance constrained programming (CCP) is an optimization framework that handles uncertainty by ensuring that constraints are satisfied with a specified probability level. This work presents GGDOpt, a novel Gradient-Guided Diffusion-based Optimization framework designed to solve chance constrained programming problems under uncertainty. This work aims to propose a framework to solve the CCP when the underlying distribution is unknown. Specifically, without requiring the exact distribution of the uncertainty, GGDOpt operates using only sampled data. It reformulates CCP as a sampling problem over the product of a data distribution and a Boltzmann distribution to effectively handle nonconvex constraints. Leveraging both first- and second-order gradient information, the method progressively injects noise through forward diffusion to convexify the feasible region and applies guided reverse sampling for optimization. This work provides some theoretical analyses for the proposed framework, such as convergence and error bounds. However, the experimental results are limited, the writing and presentation of this work need to be enhanced, and more analyses and discussions are required to be added.

**Questions:**

My questions are based on the weakness:

The third assumption in Assumption 1 seems uncommon in the existing optimization literature. Could the authors clarify whether all the assumptions used in this paper, particularly the third one, are standard and practical? Would it be possible to provide further discussion or justification for their applicability?

Could the authors include more experiments on additional machine learning applications? The current experimental settings appear limited to relatively simple scenarios, and broader empirical validation may enhance the generality of the proposed method.

The readability of Figure 1 is somewhat limited. Would the authors consider improving the caption by adding a more detailed explanation of the framework and highlighting its key components, to help readers better grasp the main insights?

The manuscript appears to lack sufficient discussion of related work. Could the authors provide a more comprehensive review and comparison with existing approaches in the field? This may help position the proposed method more clearly within the current literature.

**Ethical Concerns:**

["NO or VERY MINOR ethics concerns only"]

**Final Justification:**

My concerns have been solved.

**Limitations:**

Yes, this work discusses the limitations in Appendix A.

**Quality:**

2

**Strengths And Weaknesses:**

Strengths:

1. The key problem studied in this work is important, that is, how to  efficiently solve CCP when the underlying distribution P is unknown.

2. This work provides theoretical guarantees for the proposed framework, such as the convergence in Theorem 2 and the optimization error bound in Theorem 3.

Weaknesses:

1. The third assumption in Assumption 1 seems not to be widely used. Can you provide more discussions about whether all the used assumptions are widely used and practical?

2. In the experiment, can you conduct more experiments on additional machine learning applications? Because the author just carry out the experiments on some simple scenarios.

3. The readability of Figure 1 is poor. The authors are suggested to offer more explanation about this framework in the caption to help the readers better learn the key insights of this framework.

4. The writing and organization of this work are poor. For instance, the manuscript lacks sufficient discussion of related work. It is recommended to add a dedicated section for this purpose. Moreover, the authors should compare the proposed method with existing approaches (maybe in the related work section) to clarify the contributions of this work.

5. In Theorem 2, the convergence analyses are provided, which is appreciated by the reviewer. However, the non-asymptotic convergence rate is unknown. Can you provide some theoretical analyses or the discussions about the non-asymptotic convergence?

6. In the introduction, the discussion of the contributions is ambiguous. For instance, it is unclear why the claim “applicable to broader problem domains” is considered a **key** contribution. Since this is a theoretical work, a more appropriate key contribution might be the development of a new framework to address a novel problem.

---

> ### Author Rebuttal · Authors · 2025-07-31
>
> **Reply**
>
> We sincerely thank the reviewer for the valuable comments and constructive feedback, which have greatly helped us clarify our contributions and improve our work. We hope the following point-by-point responses address your concerns and offer a clearer understanding of our work.
>
> **Reply to Weakness 1 and Question 1:**
>
> We appreciate your feedback regarding Assumption 1. The first and second assumptions in Assumption 1 describe the forward and reverse processes of an Ornstein–Uhlenbeck (OU) process, respectively; such formulations are standard in the diffusion literature for analyzing the impact of the finite diffusion time on the convergence (e.g., [Song2020]).
>
> The third assumption imposes **a growth bound on the gradient of the objective function**. This type of regularity condition is common in the convergence analysis of stochastic optimization and sampling algorithms, particularly when studying stability and convergence under Langevin dynamics or diffusion-based methods (see, e.g., [Raginsky et al., 2017]). In practice, this assumption holds for a broad class of functions, including smooth bounded functions and quadratic objectives, which frequently arise in real-world optimization problems.
>
> **Reply to Weakness 2 and Question 2:**
>
> Following your suggestion, we have added experiments on a VaR-constrained mean–variance portfolio selection problem, which aims to minimize the risk while pursuing a targeted level of returns with probability at least $1-\rho$. Let $\boldsymbol{\mu}\in\mathbb{R}^n$ and $\boldsymbol{\Sigma}\in\mathbb{R}^{n\times n}$ denote the expectation and covariance matrix of the returns of $n$ risky assets, and $\gamma\in\mathbb{R}_+$ denote the risk aversion factor. Let $\boldsymbol{x}\in\mathbb{R}^n$ denote the allocation vector. Then this problem is formulated as follows:
>
> $$\min_{\boldsymbol{x}\in\mathbb{R}^n}\quad\gamma \boldsymbol{x}^{\top}\boldsymbol{\Sigma}\boldsymbol{x}-\boldsymbol{\mu}^{\top}\boldsymbol{x}\quad\text{s.t.}\quad\text{Prob}_{\boldsymbol{\xi}}(\boldsymbol{\xi}^{\top}\boldsymbol{x}\geq R)\geq1-\rho,$$
>
> where $R\in\mathbb{R}_+$ is a prespecified level on the return. We use 2523 daily return data of 435 stocks included in Standard \& Poor’s 500 Index between March 2006 and March 2016 and set $R=0.02\%$ and $\gamma=2$. Some results are shown below:
>
> **Table:Comparison results of the VaR-constrained mean-variance portfolio selection problem**
> |($\rho$,$n$)|Metric|MIP|ALDM|PDCA|LAG|GGDOpt(First)|GGDOpt(Second)|
> |---|---|---|---|---|---|---|---|
> |(0.05,100)|fval|-0.0951|-0.0723|-0.0917|-0.0936|-0.0904|-0.0946|
> ||time|15.58|2.418|4.602|0.9433|0.3768|0.4071|
> ||prob|0.8600|0.8666|0.9700|0.8467|0.9200|0.8933|
> |(0.05,400)|fval|-0.0874|-0.0750|-0.0814|-0.0859|-0.0827|-0.0867|
> ||time|204.2|66.68|93.42|2.7570|1.2732|1.3559|
> ||prob|0.9066|0.8308|0.9891|0.8933|0.9533|0.9267|
> |(0.1,100)|fval|-0.0951|-0.0721|-0.0856|-0.0927|-0.0915|-0.0936|
> ||time|13.31|2.388|6.258|0.9365|0.3420|0.4218|
> ||prob|0.8600|0.7633|0.9233|0.8533|0.9067|0.8667|
> |(0.1,400)|fval|-0.0874|-0.0713|-0.0826|-0.0864|-0.0829|-0.0870|
> ||time|148.6|67.95|81.95|2.7323|1.2546|1.2818|
> ||prob|0.9058|0.8158|0.9266|0.8800|0.9267|0.9133|
>
> In the above experiments, we compare our algorithm with several classical methods, including the mixed‐integer program (MIP), the augmented Lagrangian decomposition method (ALDM), the proximal difference‐of‐convex algorithm (PDCA), and the diffusion‐based Look‐Ahead Guidance (LAG) method [Guo2024]. We set $\rho=0.05,0.1$ and $n=100,400$, reporting the final‐iteration objective function value (fval), total runtime (time), and the empirical probability of the chance constraint computed over randomly sampled daily returns (prob).
>
> Table 1 shows that MIP achieves the lowest objective values but incurs the highest computational cost, as it fully exploits the data by formulating CCP as mixed integer program. LAG attains competitive objectives but requires additional back‐propagation steps for guidance. In contrast, GGDOpt well balances solution quality and efficiency, significantly reducing runtime while maintaining comparable objective values and constraint satisfaction.
>
> **Reply to Weakness 3 and Question 3:**
>
> We appreciate your suggestion on Figure 1. To help readers better understand the framework, we provide an additional explanation in the caption of Figure 1 as follows:
>
> >**Figure 1:**Overview of the GGDOpt framework.
> >(1)Generate a training set of points satisfying the chance constraint by solving a deterministic restricted problem.  (2)Train a diffusion model with classifier-free guidance to learn the score of the conditional distribution.  (3)Perform the reverse diffusion process with additional gradient guidance to sample from the product of the data distribution and the Boltzmann distribution.
>
> **Reply to Weakness 4 and Question 4**
>
> Thank you for pointing out the absence of a dedicated related‑work section. In the revised manuscript, we plan to add a standalone subsection to position our work more clearly within the literature.
>
> Compared with [Guo2024] and related methods, our work is the first, to the best of our knowledge, to use diffusion models to solve the general chance constrained problems. The key challenge here is the **lack of direct training data** corresponding to the product distribution of the objective and constraints. We address this through a dedicated data generation stage, followed by conditional training of the score. In contrast, [Guo2024] assumes access to a pre-trained unconditional diffusion model and focuses on a restricted linear-Gaussian setting. Unlike classical convex approximation approaches for CCP, our method does not require prior knowledge of the underlying distribution. Instead, we only assume access to samples from it, which makes our approach applicable to broader and more realistic settings.
>
> For clarity, we summarize the main contributions below.
>
> - **Problem reformulation with a novel paradigm**
> We reformulate the original CCP as a sampling task over the product of two distributions: an unknown data distribution (implicitly defined by the constraint) and a Boltzmann distribution induced by the objective function. This perspective enables us to transform the challenging chance constrained optimization problem into a generative modeling problem amenable to diffusion-based techniques.
> - **Feasibility-aware data generation**
> To generate high-quality training data that approximately satisfies the chance constraint, we design a deterministic restricted problem by relaxing the probability constraint and solving it using standard optimization techniques. The solutions are used to guide the training of the conditional diffusion model, effectively capturing the geometry of the feasible region without requiring explicit knowledge of the underlying uncertainty distribution.
> - **Efficient gradient guidance**
> To sample from the product distribution, we develop a gradient-guided reverse process. Our guidance terms are derived in closed form based on the structure of the product distribution and do not require backpropagation through the neural network. This is in sharp contrast to [Guo2024], where the guidance involves differentiating the posterior mean estimate with respect to the input, resulting in significantly higher computational cost.
> - **Theoretical convergence and practical evaluation**
> Regarding the sampling process as a reverse time stochastic differential equation, GGDOpt is shown to generate asymptotically optimal solutions. A practical error bound is also provided with two components: the limited time length error and limited inverse temperature error.
>
> **Reply to Weakness 5:**
>
> Thanks for your recognition of Theorem 2 and its convergence guarantee. As you correctly noted, Theorem 3 provides finite-time and finite-temperature error bounds between the samples from the reverse process and the minimizer $\boldsymbol{x}^*$ of the objective $f(\boldsymbol{x})$ within an approximated feasible region $D_{\rho}$. Here we give a corollary of Theorem 3 to present some non-asymptotic upper bounds.
>
> Corollary (Informal): For sufficiently large $\beta>0$, with the assumption in Theorem 3, we have that $\mathbb{E}[f(\tilde{\boldsymbol{x}}_T)]-f(\boldsymbol{x}^{*})]\leq O(\frac{1}{\sqrt{T}})$.
>
> It should be noted that the above corollary is a preliminary convergence result requiring no additional assumptions and similar to the Langevin algorithm for solving nonconvex problems [Ma2019].
>
> **Reply to Weakness 6:**
>
> We agree that the primary contribution of our work is the GGDOpt framework for solving chance constrained problems under unknown distributions. In the revised manuscript, we will refine the contribution statements—particularly in the introduction—to make this emphasis clearer and to guide readers better understanding our concerns and contributions.
>
> Once again, we thank the reviewer for the thorough evaluation and thoughtful suggestions. We hope this detailed response clarifies the motivation, methodology, and contributions of our work. Your feedback has been invaluable in strengthening both the presentation and the scope of the paper.
>
> The corresponding references are listed in order:
>
> 1.Song, Yang, et al. Score-based generative modeling through stochastic differential equations. arXiv preprint arXiv:2011.13456, 2020.
>
> 2.[Raginsky2017]Raginsky, Maxim, et al. Non-convex learning via stochastic gradient langevin dynamics: a nonasymptotic analysis. Conference on Learning Theory. PMLR, 2017.
>
> 3.[Ma2019]Ma, Yi-An, et al. Sampling can be faster than optimization. Proceedings of the National Academy of Sciences 116.42, 2019.
>
> 4.[Guo2024]Guo, Yingqing, et al. Gradient guidance for diffusion models: An optimization perspective. Advances in Neural Information Processing Systems, 2024.

---

> > ### Comment · Reviewer_2s5s · 2025-08-04
> >
> > Thanks for the rebuttals. My concerns have been addressed. I maintain the score.

---

### Note · Authors · 2025-08-12

We thank all reviewers and the AC for their careful reading and constructive comments. Our work proposes a novel Gradient-Guided Diffusion-based Optimization (GGDOpt) framework for chance constrained programming (CCP). To our knowledge, this is the first work to apply diffusion models to general CCP. Unlike classical convex approximation approaches, GGDOpt does not require **prior knowledge of the underlying uncertainty distribution**. Compared to the key reference [Guo2024], the critical challenge we address is **the absence of direct training data** from the product distribution of the objective and constraints.

**Key strengths and novelty:**

**Novel reformulation:**  We cast CCP as sampling from the product of an unknown feasible-set distribution defined by the chance constraint and a Boltzmann distribution induced by the objective. This reformulation enables the use of diffusion-based generative modeling for optimization.

**Feasibility-aware data generation:** We construct a deterministic reformulation of the chance constraint and solve it by using standard optimization to generate high-quality data, which achieves a good trade-off between data generation costs and solution quality.

**Efficient gradient guidance:** Our reverse process guidance is derived analytically from the product distribution, avoiding costly backpropagation through the network as in [Guo2024], and thus reducing computational overhead.

**Theoretical convergence:** Viewing sampling as a reverse-time SDE, we prove the convergence to asymptotically optimal solutions, with an explicit error bound from a finite time and inverse temperature.

**Changes to be included in order to address reviewers' concerns:** We will clarify the assumptions in Theorem 2, emphasize that the bounded score error leads to support equivalence (not identical distributions), and revise the proof to highlight this. We will also detail how condition (21) can be met in practice, expand the training and sample complexity analysis, and add a sufficient discussion of related works.

We appreciate the insightful feedback and believe that these clarifications and changes will significantly strengthen both the theoretical and empirical contributions of our work.

**Reference:**

[Guo2024] Guo, Yingqing, et al. Gradient guidance for diffusion models: An optimization perspective. Advances in Neural Information Processing Systems, 2024.

---

### Decision · Program_Chairs · 2025-09-17

**Decision:**

Accept (poster)

**Comment:**

This paper studies chance-constrained optimization problems under unknown distributions and proposes a novel optimization framework leveraging diffusion models to address them. The submission generated a constructive and detailed discussion where reviewers acknowledged both strengths and weaknesses. Concerns were raised about poor readability in some sections, missing discussion of related work, and the absence of experiments on machine learning-related applications. Nonetheless, the overall impression was positive. The paper addresses an important problem and provides a sound theoretical analysis. The proposed framework is original in several aspects, as it reformulates the chance-constrained program as a sampling task over a product of two distributions, the unknown data distribution (supported on a nonconvex set) and a Boltzmann distribution defined by the objective function. The numerical results are promising and suggest potential for broader influence. Importantly, Reviewer u5CA noted that certain results from [Pidstrigach2022] were misapplied, but this issue was clarified during the discussion. A more critical remaining concern is the reliance on condition (21). It is important that the authors revise the proofs and presentation to clearly indicate where this strong assumption is required, and to expand the discussion of related literature accordingly.